# CRITICAL POINTS IN QUANTUM GENERATIVE MODELS

**Eric R. Anschuetz**
MIT Center for Theoretical Physics
Cambridge, MA 02139, USA
`eans@mit.edu`

## ABSTRACT

One of the most important properties of neural networks is the clustering of local minima of the loss function near the global minimum, enabling efficient training. Though generative models implemented on quantum computers are known to be more expressive than their traditional counterparts, it has empirically been observed that these models experience a transition in the quality of their local minima. Namely, below some critical number of parameters, all local minima are far from the global minimum in function value; above this critical parameter count, all local minima are good approximators of the global minimum. Furthermore, for a certain class of quantum generative models, this transition has empirically been observed to occur at parameter counts exponentially large in the problem size, meaning practical training of these models is out of reach. Here, we give the first proof of this transition in trainability, specializing to this latter class of quantum generative models. We use techniques inspired by those used to study the loss landscapes of classical neural networks. We also verify that our analytic results hold experimentally even at modest model sizes.

## 1 INTRODUCTION

### 1.1 MOTIVATION

One of the great successes of neural networks is the efficiency at which they are trained via gradient-based methods. Though training algorithms often involve the optimization of complicated, non-convex, high-dimensional functions, training via gradient descent in many contexts manages to converge to local minima that are good approximations of the global minimum in loss function value. This phenomenon has begun to be understood in the context of random matrix theory, particularly when applied to dense classifiers (Choromanska et al., 2015; Chaudhari & Soatto, 2017).

Quantum generative models hold great promise in their ability to sample from probability distributions out of the reach of classical models (Arute et al., 2019; Gao et al., 2021). Though deep quantum generative models are believed to be difficult to train due to vanishing gradients (McClean et al., 2018; Cerezo et al., 2021; Marrero et al., 2020), the situation for shallow models is murkier. Some shallow quantum models have empirically shown great promise in being trainable (Wiersema et al., 2020; Kim et al., 2020; Kim & Oz, 2021), while others have empirically been shown to suffer from poor distributions of local minima (Kiani et al., 2020; Campos et al., 2021). Numerically, all of these models have been seen to experience a phase transition in trainability: below some critical depth, local minima are poor approximators of the global minimum. Above this critical depth, they are good approximators. This transition has been poorly understood analytically, as typically the distribution of local minima monotonically improves as the size of the model increases (Choromanska et al., 2015; Chaudhari & Soatto, 2017).

### 1.2 OUR CONTRIBUTIONS

In this work, we are the first to analytically show the presence of a computational phase transition in the training of a certain class of quantum generative models. To achieve this, we first show that this class of randomized quantum models is approximated in distribution by a Wishart random field

on the hypertorus. We are then able to use techniques from Morse theory to exactly calculate the distribution of local minima (and general critical points) of this random field. Finally, we analyze this distribution in the limit of large model size, and analytically show the presence of this trainability phase transition. Roughly, we show that in this limit the expected density of local minima for a model with $p$ parameters and Hilbert space dimension $\sim m$ (exponential in the problem size) at loss value $0 \leq E \leq \frac{1}{2}$ follows a generalization of the beta distribution (Gordy, 1998):

$$\mathbb{E}\left[\mathrm{Crt}_0\left(E\right)\right] \sim \mathrm{e}^{-mE} E^{m-\frac{p}{2}} \left(1-2E\right)^p . \tag{1}$$

This distribution experiences a transition in behavior at $p = 2m$: when $p < 2m$, local minima are exponentially concentrated (i.e. with width $m^{-1}$) far away from the global minimum $E = 0$, implying poor optimization performance in this regime. When $p \geq 2m$, this distribution is exponentially concentrated at $E = 0$, implying good optimization performance. We also verify our results numerically, demonstrating this concentration of minima even at small problem sizes.

For the class of quantum generative models we consider, our results mirror the empirical results of Kiani et al. (2020); Campos et al. (2021) in that only unreasonably overparameterized quantum models have good local minima. Though these results are pessimistic, we emphasize here that our results only apply to a certain class of quantum generative models. We are also able to give a heuristic explanation based on our proof techniques as to how one may be able to construct models of a reasonable size that are still trainable at the expense of computational overhead in implementing the model, as seen empirically in Wiersema et al. (2020); Kim et al. (2020); Kim & Oz (2021).

## 2 PRELIMINARIES

### 2.1 QUANTUM GENERATIVE MODELS

A quantum system of size $n$ is naturally represented by a *quantum state*, which is a normalized vector $|\psi\rangle \in \mathbb{C}^{2^n}$. Here, we use the typical physics notation $|\cdot\rangle$ to denote a vector, instead of (say) $\boldsymbol{\psi}$, when we are describing a quantum state. A quantum state in $\mathbb{C}^{2^n}$ can be considered a generalization of probability distributions over $2^n$ states (i.e. over states described by $n$ bits), where the norm squared of entries of $|\psi\rangle$ give the distribution. The task in quantum generative modeling is to prepare a state $|\psi\rangle$ that is close under some metric to a given target state $|\psi_{\text{target}}\rangle$.

As operations that map probability distributions to probability distributions are naturally described by stochastic matrices, operations that map quantum states to quantum states are naturally described by unitary matrices; equivalently, they are described by the matrix exponentials of $2^n \times 2^n$ skew-Hermitian matrices. Multiple quantum operations can then be described by the sequential matrix multiplication of various matrix exponentials. It is then apparent that one can construct a quantum generative model by parameterizing these matrix exponentials, giving rise to the layered structure:

$$|\boldsymbol{\theta}\rangle \equiv \prod_{i=1}^{q} U_i\left(\boldsymbol{\theta}\right) |\psi_0\rangle \equiv \prod_{i=1}^{q} \mathrm{e}^{-\mathrm{i}\theta_i Q_i} |\psi_0\rangle , \tag{2}$$

where here $q$ is the depth of the model and $Q_i$ are fixed Hermitian matrices. Typically, $q$ is polynomial in $n$, i.e. logarithmic in the dimension of the initial state vector $|\psi_0\rangle$. In most quantum generative models, various $\theta_i$ are completely dependent, e.g. $\theta_i = \theta_{i+5}$ for all $i$ (Romero et al., 2018). For simplicity, we assume throughout this work that each independent parameter appears a constant number $r$ times in the model, and the total number of independent parameters is given by $p = q/r$. Though the linear nature of the model may be surprising, this model structure (for large enough depth $q$) is known to be a universal approximator of all (pure) quantum states, even for a fixed number of allowed $Q_i$ (Solovay, 1995; Kitaev, 1997).

For completely general (i.e. dense) $Q_i$ in equation 2, the model is not efficient to implement on a quantum computer. Thus, due to their efficiency in physical implementation, these $Q_i$ are typically taken to be members of the *n-qubit Pauli group* $\mathbb{P}_n$, which is a generating subset of the additive group of all $n \times n$ Hermitian matrices. The Pauli group is also convenient to study analytically, as is the normalizer of the group (called the *Clifford group*). The assumption that each $Q_i$ is a Pauli operator also allows us to use a single parameter $\theta_i$ for each layer of the model without loss of generality; in principle, more parameters can describe each layer by parameterizing sparse $Q_i$. However, as the

Pauli group generates Hermitian matrices, this is a special case of the class of models we consider here (at the expense of larger $q$ and new dependencies among the $\theta_i$).

In quantum generative modeling, we are often not given the target state $|\psi_{\text{target}}\rangle$ directly. Instead, we are in many cases given a $2^n \times 2^n$ Hermitian matrix $H$ (called the *problem Hamiltonian*) where $|\psi_{\text{target}}\rangle$ is the eigenvector associated with the smallest eigenvalue of $H$ (called the *ground state*). In this formulation, optimization proceeds via the minimization of:

$$F_{\text{VQA}}(\boldsymbol{\theta}) = \langle \boldsymbol{\theta}| H |\boldsymbol{\theta}\rangle. \tag{3}$$

In physics notation, $\langle\boldsymbol{\theta}|$ is the conjugate transpose of the complex vector $|\boldsymbol{\theta}\rangle$. Assuming no degeneracies in the eigenspectrum of $H$ and a sufficiently expressive model, the minimizer $|\boldsymbol{\theta^*}\rangle$ of equation 3 is the ground state of $H$, up to an overall phase due to the quadratic nature of equation 3. Assuming $H$ is efficiently expressible as the weighted sum of $\mathrm{O}\left(\mathrm{poly}\left(n\right)\right)$ Pauli matrices, equation 3 and its gradients can be efficiently measured on a quantum computer (Peruzzo et al., 2014; Romero et al., 2018). This and similar formulations of quantum generative modeling with equation 3 as the loss function are called *variational quantum algorithms* (VQAs) (Peruzzo et al., 2014). Generally, the goal of these algorithms is to find the state $|\boldsymbol{\theta}\rangle$ that optimizes equation 3, up to potentially some constant additive error in loss. Though there are other formulations of quantum generative modeling, we here focus on VQAs as they do not require coherent access to data $|\psi_{\text{target}}\rangle$, which is generally believed to be difficult (Aaronson, 2015).

Typically, models in VQAs come in one of two flavors: Hamiltonian agnostic models, and Hamiltonian informed models. Hamiltonian agnostic models are constructed such that the $Q_i$ present in the model definition are independent of $H$, and are generally more efficient to implement. This is most analogous to the case in classical generative modeling, where the model structure is usually independent from the specific choice of data distribution. We prove the existence of a computational phase transition for a class of Hamiltonian agnostic models in this work. We also give some heuristic and numerical evidence that a similar transition exists for Hamiltonian informed ansatzes, but that the trainable phase is more easily reached.

## 2.2 RANDOM FIELDS ON MANIFOLDS

As in previous results on the loss landscapes of machine learning models (Choromanska et al., 2015; Chaudhari & Soatto, 2017), we will map the distribution of a randomized class of quantum generative models to a random field on a manifold. This will then enable us to use standard mathematical techniques to study the distribution of critical points of the model.

Though they can be expressed in many ways, here we will be interested in random fields of the form:

$$F_{\text{RF}}(\boldsymbol{\sigma}) \propto \sum_{i_1,\ldots,i_r,i_1',\ldots,i_r'=1}^{\Lambda} \sigma_{i_1} \ldots \sigma_{i_r} J_{i_1,\ldots,i_r,i_1',\ldots,i_r'} \sigma_{i_1'} \ldots \sigma_{i_r'}. \tag{4}$$

Here, $\boldsymbol{\sigma} \in M$ is some point on a manifold, and $\boldsymbol{J}$ is a random variable. In the context of most studies of machine learning loss landscapes, $M$ is typically the hypersphere and $\boldsymbol{J}$ a symmetric matrix of i.i.d. Gaussian random variables, i.e. a Gaussian orthogonal ensemble (GOE) matrix.

We will instead find that VQAs are naturally described as *Wishart hypertoroidal random fields* (WHRFs). For these models, the manifold $M$ is a tensor product embedding of the hypertorus into exponentially large Euclidean space; that is, points on this embedding are described by the Kronecker product:

$$\boldsymbol{\sigma} = \bigotimes_i \begin{pmatrix} \cos\left(\theta_i\right) \\ \sin\left(\theta_i\right) \end{pmatrix} \tag{5}$$

for angles $-\pi \leq \theta_i \leq \pi$. Furthermore, in these models, $\boldsymbol{J}$ is drawn from a normalized complex Wishart distribution. The complex Wishart distribution is a natural multivariate generalization of the gamma distribution, and is given by the distribution of the square of a complex Gaussian random matrix. Specifically, for $\boldsymbol{X} \in \mathbb{C}^{n \times m}$ a matrix with i.i.d. complex Gaussian columns with covariance matrix $\boldsymbol{\Sigma}$, the matrix

$$\boldsymbol{W} = \frac{1}{m}\boldsymbol{X} \cdot \boldsymbol{X}^\dagger \tag{6}$$

is normalized complex Wishart distributed with scale matrix $\boldsymbol{\Sigma}$ and $m$ degrees of freedom. We will find that the degrees of freedom $m$ will greatly affect the distribution of local minima of the WHRF, and thus also of the class of quantum generative models that we consider.

## 3 Quantum Generative Models as Random Fields

We first show that a certain randomized class of Hamiltonian agnostic VQAs can be expressed as WHRFs. This will allow us to more easily study the critical points of the model using techniques from random matrix theory. Though we leave the full statement and proof for Appendix A, we give an informal statement and discussion here.

**Theorem 1** (VQAs as WHRFs, informal)**.** *Consider the class of models*

$$|\boldsymbol{\theta}\rangle \equiv \prod_{i=1}^{q} U_i\left(\boldsymbol{\theta}\right)|\psi_0\rangle \equiv \prod_{i=1}^{q} \mathrm{e}^{-\mathrm{i}\theta_i Q_i}\left|\psi_0\right\rangle, \tag{7}$$

*where each $Q_i$ is drawn uniformly from the Pauli group $\mathbb{P}_n$ and $|\psi_0\rangle$ is the first column of a uniformly random member of the Clifford group. Let $p$ be the number of distinct $\theta_i$, and let $r = q/p$. Under reasonable assumptions on the eigenvalues of $H$ (with minimum eigenvalue $\lambda_1$ and mean eigenvalue $\overline{\lambda}$), the random loss function*

$$F_H\left(\boldsymbol{\theta}\right) = \frac{\langle\boldsymbol{\theta}|\,H\,|\boldsymbol{\theta}\rangle - \lambda_1}{\overline{\lambda} - \lambda_1} \tag{8}$$

*converges in distribution to the random field*

$$F_{WHRF}\left(\boldsymbol{\theta}\right) = \sum_{i_1,\ldots,i_r,i'_1,\ldots,i'_r=1}^{2^p} w_{i_1}\ldots w_{i_r} J_{i_1,\ldots,i_r,i'_1,\ldots,i'_r} w_{i'_1}\ldots w_{i'_r}, \tag{9}$$

*where $\boldsymbol{w}$ are points on the hypertorus $\left(S^1\right)^{\times p}$ parameterized by $\boldsymbol{\theta}$ and $\boldsymbol{J}$ is a complex Wishart random matrix normalized by its number of degrees of freedom $m$.*

Note that for convenience, we have shifted and scaled the typical VQA loss such that it is always greater than zero and independent from overall scalings of the problem Hamiltonian.

In the course of this mapping, we find that the *degrees of freedom* of the Wishart matrix $\boldsymbol{J}$ (formally a real number) is given by the ratio:

$$m \equiv \frac{\|H - \lambda_1\|_*^2}{\|H - \overline{\lambda}\|_{\mathrm{F}}^2}. \tag{10}$$

Here, $\|\cdot\|_*$ denotes the nuclear norm, and $\|\cdot\|_{\mathrm{F}}$ the Frobenius norm. Generally, this ratio is exponential in $n$, particularly when modeling the class of ground states typically represented by VQAs (Peruzzo et al., 2014; Farhi et al., 2014; Romero et al., 2018). Though we are unable to prove Theorem 1 for Hamiltonian informed models, there are heuristic reasons to believe that they are described by a similar random field with $m = \mathrm{O}\left(\mathrm{poly}\left(n\right)\right)$, as opposed to equation 10 (see Appendix A.3 and empirical evidence in Sec. 5). We will later find that a number of independent model parameters $p$ that is twice the degrees of freedom $m$ of the matrix $\boldsymbol{J}$ marks the transition from the underparameterized to the overparameterized regime of $F_{\mathrm{WHRF}}$, where the quality of local minima improves.

The general idea for showing this equivalence relies on the *path integral expansion* of the VQA loss function. Effectively, this is just a Taylor expansion of the unitary matrices composing the model, which is exact even at a finite number of terms. One can then show that terms in this expansion can be assumed independent with negligible error in distribution, and then show that the resulting random process is asymptotically a WHRF. The reasonable assumptions on the eigenvalues of $H$ are essentially just a requirement that the eigenvalues of $H$ are not "unnaturally" spread out; for the quantum states VQAs typically model, this is never the case. We give a full description of these requirements with the full proof in Appendix A.

## 4  THE LOSS LANDSCAPE OF WISHART HYPERTOROIDAL RANDOM FIELDS

### 4.1  EXACT RESULTS

Having shown that VQAs can be described as WHRFs, we now focus discussion entirely on WHRFs. Our strategy for showing the distribution of critical points of this random field will be similar to that in Auffinger et al. (2013), where similar results were shown for Gaussian spherical random fields. Namely, we will lean heavily on the *Kac–Rice formula*, which gives the expected number of critical points of a certain index at a given range of function values for random fields on manifolds. We give an informal description of the Kac–Rice formula here, with the formal version given in Appendix B.

**Lemma 1** (Kac–Rice formula (Adler & Taylor, 2009), informal). *Let $M$ be a compact, oriented manifold. Assume a random field $F(\boldsymbol{\sigma})$ on $M$ is sufficiently nice. Then, the number of critical points of index at most $k$ with $F(\boldsymbol{\sigma}) \in B$ for an open set $B \subset \mathbb{R}$ is*

$$\mathbb{E}\left[\mathrm{Crt}_k(B)\right] = \int_M \mathbb{E}\left[\left|\det\left(\boldsymbol{\nabla}^2 F(\boldsymbol{\sigma})\right)\right| \mathbf{1}\left\{F(\boldsymbol{\sigma}) \in B\right\} \mathbf{1}\left\{\iota\left(\boldsymbol{\nabla}^2 F(\boldsymbol{\sigma})\right) \le k\right\} \mid \boldsymbol{\nabla} F(\boldsymbol{\sigma}) = 0\right]$$
$$\times p_{\boldsymbol{\sigma}}\left(\boldsymbol{\nabla} F(\boldsymbol{\sigma}) = 0\right) \mathrm{d}\boldsymbol{\sigma},$$
(11)

*where $\boldsymbol{\nabla}\cdot$ is the covariant gradient, $\iota(\cdot)$ is the index of $\cdot$, $p_{\boldsymbol{\sigma}}$ is the probability density of $\boldsymbol{\nabla} F(\boldsymbol{\sigma})$ at $\boldsymbol{\sigma}$, and $\mathrm{d}\boldsymbol{\sigma}$ is the volume element on $M$.*

From Lemma 1, we see that when the joint distribution of $\boldsymbol{\nabla}^2 F$, $\boldsymbol{\nabla} F$, and $F$ is known, then the expected number of critical points with function values in an open set $B$ can be calculated. Perhaps surprisingly, as in the Gaussian case, the joint distribution of these derivatives for WHRFs is fairly simple. Once again leaving the full proof for Appendix C, we show the distribution of the Hessian conditioned to be at a critical point of function value $x$ can be described by the shifted sum of a Wishart matrix with an independent GOE matrix. Similarly, the distribution of the gradient conditioned on the function value being $x$ is given by a normal distribution.

**Lemma 2** (Hessian and gradient distributions, informal). *The scaled Hessian $m\partial_i\partial_j F_{WHRF}(\boldsymbol{w})$ conditioned on $F_{WHRF}(\boldsymbol{w}) = x$ and $\partial_k F_{WHRF}(\boldsymbol{w}) = 0$ is distributed as*

$$m\tilde{C}_{ij}(x) = -2rmx\delta_{ij} + rW_{ij} + r\sqrt{2mx}N_{ij},$$
(12)

*where $\boldsymbol{W}$ is Wishart distributed with $2m$ degrees of freedom, $\boldsymbol{N}$ GOE distributed, and they are independent. Furthermore, the scaled gradient $m\partial_k F_{WHRF}(\boldsymbol{w})$ conditioned on $F_{WHRF}(\boldsymbol{w}) = x$ is distributed as*

$$m\tilde{G}_k(x) = \sqrt{2mrx}N_k,$$
(13)

*where $N_k$ are i.i.d. standard normal distributions independent from all $W_{ij}$ and $N_{ij}$.*

With all of the pieces in place, we are able to explicitly calculate the expected distribution of local minima in WHRFs via the Kac–Rice formula (with full calculations left for Appendix C). In Sec. 5 we find empirical evidence that these results hold not only in expectation, but in distribution; we leave further analytic investigation of this to future work.

**Theorem 2** (Distribution of critical points in WHRFs, informal). *Let*

$$\mu_{\boldsymbol{C}(x)} = \frac{1}{p}\sum_{i=1}^{p}\delta\left(\lambda_i^{\boldsymbol{C}(x)}\right)$$
(14)

*be the empirical spectral measure of the random matrix*

$$\boldsymbol{C}(x) = \frac{r}{m}\left(\boldsymbol{W} + \sqrt{2mx}\boldsymbol{N}\right),$$
(15)

*where $\boldsymbol{W}$ is Wishart distributed with $2m$ degrees of freedom, $\boldsymbol{N}$ GOE distributed, and they are independent. $\lambda_i^{\boldsymbol{C}}(x)$ is the $i$th smallest eigenvalue of $\boldsymbol{C}(x)$. Then, the distribution of the expected number of critical points of index $k$ of a WHRF at a function value $E > 0$ is given by*

$$\mathbb{E}\left[\mathrm{Crt}_k(E)\right]$$
$$= \left(\frac{\pi}{r}\right)^{\frac{p}{2}}\Gamma(m)^{-1}m^{(1+\gamma)m}\mathbb{E}_{\boldsymbol{C}(E)}\left[\mathrm{e}^{p\int\ln(|\lambda-2rE|)\mathrm{d}\mu_{\boldsymbol{C}(E)}}\mathbf{1}\left\{\lambda_{k+1}^{\boldsymbol{C}(E)} \ge 2rE\right\}\right]E^{(1-\gamma)m-1}\mathrm{e}^{-mE},$$
(16)

*where*

$$\gamma = \frac{p}{2m}. \tag{17}$$

We call the parameter $\gamma$ the *overparameterization factor*. It describes the ratio between the number of parameters of the model $p$ and twice the degrees of freedom $m$ of the model. We will later find that $\gamma$ governs the phase transition between an *underparameterized phase* of the model—where local minima are far from the global minimum—and an *overparameterized phase*—where local minima are good approximators of the global minimum.

## 4.2 ASYMPTOTIC RESULTS AS $p \to \infty$

Though Theorem 2 gives the exact distribution of critical points, it is difficult to use in practice. This difficulty comes from the expectation over eigenvalues of the sum of independent Wishart and Gaussian matrices. Surprisingly, however, the eigenvalues of both Wishart and Gaussian orthogonal matrices converge to *fixed* distributions. Essentially, asymptotically in the size of the matrix, the eigenvalue distribution of all normalized Wishart matrices are the same (given by the *Marchenko–Pastur distribution*) and the eigenvalue distribution of all Gaussian orthogonal matrices are the same (given by the *Wigner semicircle distribution*).

Luckily, we can characterize the asymptotic distribution of eigenvalues of the sums of these matrices using the tools of *free probability theory*. Roughly, free probability theory is the probability theory of noncommutative random variables (e.g. random matrices). As the distribution of the sum of two random variables in commutative probability theory can be described by the convolution of the distributions of the two independent random variables, so can the *free convolution* of the distributions of two *freely independent* noncommutative random variables. Using the asymptotic free independence of Wishart and Gaussian orthogonal random variables, we are able to show that asymptotically the eigenvalue distribution of their sum weakly converges to the free convolution of a Marchenko–Pastur distribution with a semicircle distribution.

However, weak convergence is not enough; due to the exponential factor in the expectation in Theorem 2, any large deviations from the asymptotic convergence—even if they occur with exponentially vanishing probability—can potentially cause large deviations from the naive application of free probability theory. Thus, our results rely on using large deviations theory to show that to (logarithmic) leading order these deviations do not contribute to the final result. This is due to the contribution to the expectation from the deviations being dominated by what is predicted by free probability theory. These results can be summarized via the following theorem (proved in Appendix D):

**Theorem 3** (Logarithmic asymptotics of the local minima distribution, informal). *Let $\mathrm{d}\mu_E^*$ be the free convolution of a scaled Marchenko–Pastur and scaled Wigner semicircle distribution, with $\lambda_{E,1}^*$ the infimum of its support. Let $p, m \gg 1$ with $\frac{p}{2m} = \gamma = \mathrm{O}(1)$. Then, the expected distribution of local minima of a WHRF at a fixed function value $E > 0$ is given by*

$$\begin{aligned}
\frac{1}{p} \ln \left( \mathbb{E}\left[ \mathrm{Crt}_0\left( E \right) \right] \right) = {} & \frac{1}{2} \ln \left( \frac{\pi q}{2\gamma} \right) + \frac{1}{2\gamma}\left( 1 - E \right) + \frac{1}{2}\left( \gamma^{-1} - 1 \right) \ln \left( E \right) \\
& + \int \ln \left( \left| \frac{\lambda}{r} - 2E \right| \mathbf{1}\left\{ \frac{\lambda_{E,1}^*}{r} \geq 2E \right\} \right) \mathrm{d}\mu_E^* + \mathrm{o}\left( 1 \right).
\end{aligned} \tag{18}$$

Note that, though we only prove the asymptotic distribution of local minima in Theorem 3, we expect similar theorems to also hold for critical points of constant index $k$ (taking $\lambda_{E,1}^* \mapsto \lambda_{E,k}^*$ in the integrand). The only difference in the derivation is the exact form of the large deviations of the $k$th smallest eigenvalue of $\boldsymbol{C}\left( x \right)$. This is similar to the case in Gaussian hyperspherical random fields, which are often used to model neural network loss functions (Auffinger et al., 2013; Choromanska et al., 2015; Chaudhari & Soatto, 2017).

## 4.3 DISCUSSION OF THE CRITICAL POINT DISTRIBUTION

Let us now discuss the implications of Theorem 3. Note first that the rescaled logarithmic number of critical points diverges when $q \to \infty$. Following the derivation closely, one finds that this is

due to an exponentially suppressed (when $m$ is exponential in $n$) gradient. We believe that this is a manifestation of the "barren plateau" phenomenon, where for many deep VQA models it can be shown that there is an exponentially vanishing variance of the gradient over the loss landscape (McClean et al., 2018; Cerezo et al., 2021; Marrero et al., 2020). This interpretation suggests that these barren plateau regions are filled with many small "bumps" that are exponentially shallow. Furthermore, note that this class of random fields exhibits banded behavior in the eigenvalues. That is, local minima only exist in the band $0 \leq E \leq E_0$, where $E_0$ is the solution to

$$\lambda_{E_0,1}^* = 2rE_0. \tag{19}$$

This banded behavior is similar to that in the Gaussian spherical case. We will see, however, that this does not give necessarily good guarantees on the distribution of local minima. This is due to $E_0$ being generally far from 0 when $\gamma < 1$ as $p, m \to \infty$. To illustrate this, we focus now on two cases: $p \geq 2m$ (the *overparameterized regime*) and $p \ll m$ (the *underparameterized regime*).

### 4.3.1 $p \geq 2m$

First, let us consider when $p \geq 2m$, i.e. $\gamma \geq 1$. In this limit, the Wishart term of $C$ is low-rank, and $\mu_E^*$ has support on eigenvalues $\leq 0$ for all $E \geq 0$. Therefore, the condition $\lambda_{E,1}^* \geq 2Er$ is never satisfied, and to leading order in $p$ there are no local minima at a function value $E > 0$. That is, all local minima are global minima in the $p \to \infty$ limit when $\gamma \geq 1$. Though the choice of model is slightly different, we suspect that a related phenomenon may be what gives rise to the phase transition in training numerically observed in Kiani et al. (2020); Wiersema et al. (2020); Kim et al. (2020); Kim & Oz (2021); Campos et al. (2021).

### 4.3.2 $p \ll m$

When the number of distinct parameters $p$ is poly $(n)$ and considering a physically relevant problem Hamiltonian such that the number of degrees of freedom $m$ is exp $(n)$, we have that $p \ll m$ (i.e. $\gamma \ll 1$) for large $n$. In this limit, the spectral distribution $\mu_E^*$ is dominated by the Wishart term of $C$, as its eigenvalues are O $(1)$ while the eigenvalues of the GOE term are O $(\sqrt{\gamma})$. Furthermore, the Marchenko–Pastur distribution in this limit only has support at $\lambda = 1 + \mathrm{O}(\sqrt{\gamma})$. Therefore, the expected number of local minima at a function value $E$ will be proportional to

$$\mathbb{E}\left[\mathrm{Crt}_0(E)\right] \propto \mathrm{e}^{-mE + \mathrm{o}(p)} E^{m - \frac{p}{2}} \left(1 - 2E + \mathrm{O}(\sqrt{\gamma})\right)^p \mathbf{1}\left\{0 \leq E + \mathrm{O}(\sqrt{\gamma}) \leq \frac{1}{2}\right\}. \tag{20}$$

In particular, up to shifts on the order of $\sqrt{\gamma}$, the distribution of local minima is roughly that of a compound confluent hypergeometric (CCH) distribution (Gordy, 1998). The CCH distribution can be considered a generalization of the beta distribution, and for our parameters has mean on the order of $\frac{1}{2} - \gamma$ and standard deviation on the order of $m^{-1}$. Restoring the overall scaling in equation 8, this implies that in this limit the local minima of the variational loss function exponentially concentrate (in expectation) near half the mean eigenvalue of $H - \lambda_1$ instead of the smallest eigenvalue. Even worse, the CCH distributed form of the local minima implies that, even when beginning at an initial function value well below half of the mean eigenvalue of $H - \lambda_1$, the found loss will only improve by a fraction of the initial function value before the optimization algorithm finds a local minimum. This is insufficient to find the optimal loss to constant additive error when beginning training at a random point, as is often the goal in VQAs (Peruzzo et al., 2014). Empirically, we find that this occurs not just in expectation but also for individual model instances in Sec. 5.

## 5 NUMERICAL EXPERIMENTS

We now test our analytic predictions using numerical simulations. First, we investigate the empirical performance of the class of randomized models we study theoretically, and give numerical evidence of things we were unable to prove. Then, we give numerical evidence that, for models dependent on the objective Hamiltonian, the effective degrees of freedom parameter $m$ can be much smaller than predicted. In all cases, we numerically test the predictions of our results by modeling the ground state of the 1D $n$ site *spinless Fermi–Hubbard Hamiltonian* (Negele & Orland, 1998) at half filling. Here, we take units such that the mean eigenvalue of the considered Hamiltonian (minus its smallest eigenvalue) is $E = 1$. We give further details of our numerical simulations in Appendix E.

## 5.1 EMPIRICAL PERFORMANCE OF RANDOM ANSATZES

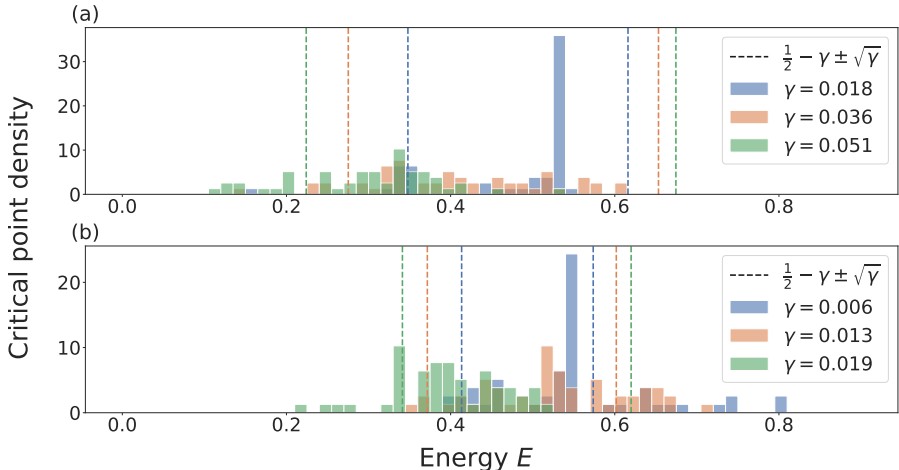

Figure 1: Here we plot the distribution of found local minima found after $52$ separate training instances using the randomized model on (a) $2^n = 64$- and (b) $2^n = 256$-dimensional models. Dashed lines denote the predicted region local minima will lie. Note the clustering of local minima at a finite function value when $\gamma \ll 1$.

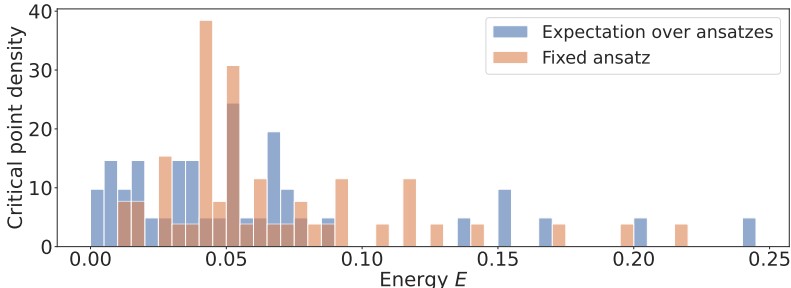

Figure 2: Here we plot the distribution of found local minima found after $52$ separate training instances using the randomized model, with $p = 48$ and $2^n = 64$ model dimension. For even a small model size, qualitatively the expected distribution of critical points and the distribution of critical points for a fixed random ansatz are in agreement.

First, we analyzed the performance of a VQA on this loss function via the random model construction procedure defined in Theorem 1. Previous numerical results on related Hamiltonian agnostic ansatzes have already shown the concentration of local minima far away in loss value from the global minimum below some degrees-of-freedom transition, and concentration at the global minimum above this transition (Kiani et al., 2020; Campos et al., 2021). Here, we tracked where our analysis predicts the local minima to lie as a function of $\gamma$ for $\gamma \ll 1$, up to deviations on the order of $\sqrt{\gamma}$ that arise from numerically considering the problem at finite size (as discussed in Sec. 4.3.2).

Concretely, for a given training instance and depth $q = p$, we generated an ansatz $|\boldsymbol{\theta}\rangle$ composed of $p$ layers of Pauli rotations, where each Pauli rotation was chosen uniformly from all nonidentity Pauli matrices on $n$ qubits. The numbers of model layers we consider are typical of current physical implementations of Hamiltonian agnostic VQAs (Kandala et al., 2017). A summary of the normalized distribution of found local minima for the randomized model with model dimension $2^n = 64, 256$ is given in Figure 1, along with the predicted region in which all local minima should lie in the $p \to \infty$ limit as discussed in Theorem 3. See Appendix E for details on how this distribution was generated.

We see that almost all found local minima lie within the predicted region, even at small $p, n$. In particular, for small $\gamma$, the distribution of local minima is almost entirely localized within $\sqrt{\gamma}$ of the

predicted $\frac{1}{2} - \gamma$ (in units of the mean eigenvalue of $H - \lambda_1$). Finally, we numerically observe that the distribution of local minima are qualitatively similar in expectation and for a single choice of random model in Figure 2.

## 5.2 EMPIRICAL PERFORMANCE OF A HAMILTONIAN INFORMED MODEL

Previous numerical results (Wiersema et al., 2020) on VQAs have shown that only a moderate number of model parameters suffices for efficient training when using a Hamiltonian informed model. As discussed in Sec. 3, we believe this is due to this class of models effectively limiting the degrees of freedom $m$ of the associated WHRF model; to test this, we performed more numerical experiments using a Hamiltonian informed ansatz. We once again tracked where our analysis predicts the local minima to lie as a function of $\gamma$ for $\gamma \ll 1$, up to deviations on the order of $\sqrt{\gamma}$.

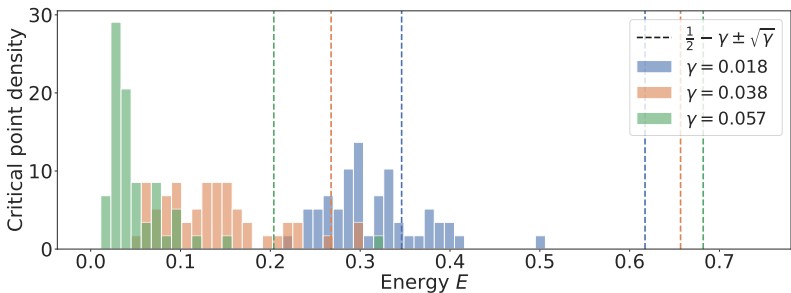

Figure 3: Here we plot the distribution of found local minima after 52 separate training instances using a Hamiltonian informed model. Dashed lines denote the predicted region local minima will lie. We see that the predicted region is overly pessimistic. We believe that this is due to the Hamiltonian informed model lowering the effective degrees of freedom $m$ of the WHRF instance the ansatz maps to; see Sec. 3.

We show the empirical distribution of local minima in Fig. 3 for $2^n = 256$, along with the predicted region local minima should lie as discussed in Sec. 4.3.2. The predicted local minima distribution is overly pessimistic (particularly at larger $p$). We suspect this is due to the fact that the ansatz is constructed in a way that minimizes the effective degrees of freedom of the WHRF $m$ such that $\gamma$ is close to 1 for smaller $p$ than is predicted analytically.

## 6 CONCLUSION

Though variational quantum algorithms are perhaps the most promising way to use the error-prone quantum devices of today for practical computational tasks, there are many caveats with regard to their trainability. In particular, previous work has shown that utilizing deep quantum models that are independent of the problem Hamiltonian can introduce a vanishing gradient phenomenon where, though the model is expressive enough to capture the ground state of interest, in practice optimizing the loss function is infeasible (McClean et al., 2018; Cerezo et al., 2021; Marrero et al., 2020). We extended these results by showing a particular class of random models independent of the problem instance not only can exhibit these vanishing gradients at large depth, but also has a concentration of local minima near the mean eigenvalue of the objective Hamiltonian. This is in contrast to the case in traditional neural networks, where even generic model structure tends to lead to a concentration of local minima in a band near the global minimum of the loss function.

Though our results may not seem encouraging for quantum generative models, we emphasize that we expect our analytic results to hold only when the model is independent of the problem Hamiltonian. Indeed, we found empirically (and heuristically) good performance for a particular Hamiltonian informed ansatz, where our analytic results seem much too pessimistic. In principle, this new way of thinking about variational quantum algorithms may inform future quantum generative model design; we leave for future work the study of how various model choices may impact the distribution of critical points of the loss function positively, and how practical considerations such as noisy model implementations may play a role.

REPRODUCIBILITY STATEMENT

The full statement of Theorem 1, along with all of its assumptions and its proof, is given in Appendix A. Similarly, Lemma 2 and Theorem 2 are fully stated and proved in Appendix C. These results rely on the Kac–Rice formula, informally stated as Lemma 1; for completeness, we give the full assumptions of this formula in Appendix B, and there also show the assumptions are met for the model we consider. Furthermore, the full assumptions and proof of Theorem 3 is given in Appendix D, along with the proofs of supplementary lemmas needed to prove the statement. Finally, full details of our numerical simulations and how to reproduce the results are given in Appendix E.

ACKNOWLEDGMENTS

We are grateful to Tongyang Li, John Napp, and Aram Harrow for insightful discussion and helpful remarks regarding a draft of this work. E.R.A. is supported by the National Science Foundation Graduate Research Fellowship Program under Grant No. 4000063445, and a Lester Wolfe Fellowship and the Henry W. Kendall Fellowship Fund from M.I.T.

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

# A  VARIATIONAL QUANTUM ALGORITHMS AS RANDOM FIELDS

## A.1  VARIATIONAL QUANTUM ALGORITHMS

Variational quantum algorithms (VQAs) are a class of quantum generative model where one expresses the solution of some problem as the smallest eigenvalue and its corresponding eigenvector (typically called the *ground state*) of an objective Hermitian matrix $H$. Given a choice of generative model—often called an *ansatz* in the quantum algorithms literature:

$$|\boldsymbol{\theta}\rangle = \prod_{i=1}^{q} U_i(\theta_i) |\psi_0\rangle \tag{21}$$

that for some choice $\boldsymbol{\theta}$ closely approximates the ground state of $H$, the solution is encoded as the minimum of the loss function

$$\tilde{F}(\boldsymbol{\theta}) = \langle \boldsymbol{\theta}| H |\boldsymbol{\theta}\rangle. \tag{22}$$

This loss function can be computed on a quantum computer efficiently, under some conditions on the matrix $H$. For simplicity of analysis, throughout this paper we will consider the loss function

$$F(\boldsymbol{\theta}) = \frac{\langle \boldsymbol{\theta}| H |\boldsymbol{\theta}\rangle - \lambda_1}{\bar{\lambda} - \lambda_1}, \tag{23}$$

where $\lambda_1$ is the smallest eigenvalue of $H$; this has the same loss landscape as equation 22, but is minimized at $F = 0$ (assuming a sufficiently expressive $|\boldsymbol{\theta}\rangle$) and is normalized by the mean eigenvalue of $H - \lambda_1$. In equation 21, $q$ is referred to as the *depth* of the circuit, and the initial vector (i.e. quantum state) $|\psi_0\rangle \in \mathbb{C}^{2^n}$ is fixed throughout the optimization procedure. Different choices of $U_i$ constitute different choices of ansatz for the ground state of $H$.

Ansatz design choice generally falls in one of two categories: Hamiltonian informed ansatzes, and Hamiltonian agnostic ansatzes. Examples of Hamiltonian informed ansatzes include the chemistry-inspired unitary coupled cluster ansatz (Peruzzo et al., 2014) and the adiabatically inspired quantum approximate optimization algorithm (QAOA) ansatz (Farhi et al., 2014), known outside of the context of combinitarial optimization as the Hamiltonian variational ansatz (HVA) (Wecker et al., 2015). These ansatzes depend solely on the problem objective Hamiltonian $H$, and are usually physically motivated ansatzes which, in some limit, have convergence guarantees. Hamiltonian agnostic ansatzes, conversely, depend solely on the hardware the VQA is run on, and not at all on the problem objective $H$. This class of ansatzes includes the hardware-efficient ansatz (Kandala et al., 2017). These ansatzes are designed to eke out as much depth as possible in the objective ansatz $|\boldsymbol{\theta}\rangle$ by using $U_i$ that can be easily implemented on the given quantum device.

Though hardware-efficient ansatzes generally can be run at larger depth $q$ than Hamiltonian informed ansatzes, the very generic nature of the ansatz circuit means this class of ansatz is more difficult to train, often encountering barren plateaus in the optimization landscape that are difficult to escape from when $q$ is large (McClean et al., 2018; Cerezo et al., 2021; Marrero et al., 2020). Heuristically, this can be understood as Hamiltonian agnostic objective functions being so expressive that it must explore essentially all of Hilbert space to find a local minimum, exponentially suppressing the gradients of the loss function (Sim et al., 2019).

In this work we consider a class of ansatzes that, like the hardware-efficient ansatz, is independent of the problem instance. In particular, we consider random parameterized ansatzes of the form:

$$U_i \equiv e^{-i\theta_i Q_i} \tag{24}$$

for Pauli operators $Q_i$, where each $Q_i$ is drawn uniformly and independently from the $n$-qubit Pauli operators. Throughout this paper, we will use $q$ to denote the total number of Pauli rotations in $|\boldsymbol{\theta}\rangle$ as in equation 21, $p$ to denote the total number of independent parameters $\theta_i$, and $r_i$ to denote the number of Pauli rotations governed by a single independent parameter $\theta_i$. For simplicity, we will assume $r_i = r_j \equiv r$ for all $i, j$, and thus take

$$r \equiv \frac{q}{p} \tag{25}$$

to be a natural number.

A.2 MAPPING VARIATIONAL QUANTUM ALGORITHMS TO RANDOM WISHART FIELDS

With the background of VQAs in place, we will now show the asymptotic (weak) equivalence of VQAs with the random choice of ansatz described in Appendix A.1 to Wishart random fields. Throughout this section, we will consider a problem Hamiltonian $H$ on $n$ qubits, with ground state energy $\lambda_1$ and mean eigenvalue $\overline{\lambda}$. We also define the *degrees of freedom* parameter

$$m \equiv \frac{\|H - \lambda_1\|_*^2}{\|H - \overline{\lambda}\|_F^2}, \tag{26}$$

whose interpretation will be discussed in Appendix A.3. Twice the degrees of freedom parameter $m$ will turn out to govern the location of the transition from the underparameterized to the over-parameterized regime (see Appendix C), and for physically relevant Hamiltonians is expected to be exponential in $n$ (see Appendix A.3). We will also consider the Pauli decomposition of the nontrivial part of $H$:

$$H - \overline{\lambda} = \sum_{i=1}^{A} \alpha_i R_i, \tag{27}$$

where $A$ is the number of terms in the Pauli decomposition and $\boldsymbol{\alpha}$ the Pauli coefficients.

We begin by showing the convergence of a class of randomized VQAs to a weighted sum of Wishart random fields at a rate $\gtrsim \log(n)$; the seemingly arbitrary shifts by the mean eigenvalue $\overline{\lambda}$ and the ground state energy $\lambda_1$ here will aid in future discussion, when we approximate the weighted sum of Wishart random fields with a single random field. The wide variety of assumptions will be discussed in detail in Appendix A.3.

**Theorem 4** (VQAs as RFs). *Let $|\psi_0\rangle$ be an arbitrary stabilizer state (e.g. a computational basis state) on $n$ qubits. Fix a sequence of $q$ angles $\theta_i \in [-\pi, \pi]$ such that each $\theta_i$ is present $r$ times in the sequence. We let $p = \frac{q}{r}$ denote the number of distinct parameters. Select an ansatz*

$$|\boldsymbol{\theta}\rangle \equiv \prod_{i=1}^{q} U_i(\boldsymbol{\theta}) |\psi_0\rangle \equiv \prod_{i=1}^{q} e^{\mp i\theta_i Q_i} C |\psi_0\rangle \tag{28}$$

*by independently at random drawing each $\pm Q_i$ uniformly from the $n$-qubit Pauli group $\mathbb{P}_n$ and $C$ from the $n$-qubit Clifford group $\mathbb{C}_n$. Consider the scaled and shifted*

$$\tilde{H} \equiv \frac{H - \overline{\lambda}}{\overline{\lambda} - \lambda_1} = \frac{H - \overline{\lambda}}{2^{-n} \|H - \lambda_1\|_*}, \tag{29}$$

*where $\|\cdot\|_*$ is the nuclear norm. Then, the random variational objective function*

$$F_{VQA}(\boldsymbol{\theta}) = \frac{\langle \boldsymbol{\theta}| H |\boldsymbol{\theta}\rangle - \lambda_1}{\overline{\lambda} - \lambda_1} = \frac{\langle \boldsymbol{\theta}| H |\boldsymbol{\theta}\rangle - \lambda_1}{2^{-n} \|H - \lambda_1\|_*} \tag{30}$$

*has first two moments exponentially close in $n$ as $n \to \infty$ to those of*

$$F_{XHX}(\boldsymbol{w}) = 2^{-n} \left( \bigotimes_{i=1}^{p} \boldsymbol{w_i^\mathsf{T}} \right)^{\otimes r} \cdot \boldsymbol{X} \cdot \tilde{\boldsymbol{H}} \cdot \boldsymbol{X}^\dagger \cdot \left( \bigotimes_{i=p}^{1} \boldsymbol{w_i} \right)^{\otimes r} + 1, \tag{31}$$

*where $\boldsymbol{w_i}$ are points on the circle parameterized by $\theta_i$ and $\boldsymbol{X}$ is a matrix of i.i.d. complex standard jointly normal random variables. Furthermore, assuming*

$$\frac{\|\boldsymbol{\alpha}\|_\infty}{\overline{\lambda} - \lambda_1} \leq f(n)^{-1} \tag{32}$$

*for some $f(n) = \Omega(1)$, their distributions are bounded in Lévy distance by $\tilde{O}\left( \left( \frac{\lg(A)f(n)n}{A} \right)^{-1} \right)$.*

*Proof.* The Feynman path integral representation (i.e. the exact Taylor expansion of the matrix exponentials using the fact that Pauli operators square to the identity) of the objective function equation 30 is of the form

$$F_{\text{VQA}} = \sum_{\boldsymbol{\gamma}, \boldsymbol{\gamma'} \in \{0,1\}^{\times q}} w_{\boldsymbol{\gamma'}}^\dagger w_{\boldsymbol{\gamma}} \langle \psi_0| C^\dagger Q_{\boldsymbol{\gamma'}}^\dagger \tilde{H} Q_{\boldsymbol{\gamma}} C |\psi_0\rangle + 1, \tag{33}$$

where $\boldsymbol{\gamma}$ labels a term in the path integral expansion of $U$,

$$w_{\boldsymbol{\gamma}} \equiv \prod_{i=1}^{q} \left\{ \begin{array}{ll} \cos\left(\theta_i\right), & \text{if } \gamma_i = 0 \\ \sin\left(\theta_i\right), & \text{if } \gamma_i = 1 \end{array} \right. \tag{34}$$

is the amplitude, and

$$Q_{\boldsymbol{\gamma}} \equiv (-\mathrm{i})^{\|\boldsymbol{\gamma}\|_0} \prod_{i=1}^{q} Q_i. \tag{35}$$

We can rewrite the Feynman path integral as

$$F_{\text{VQA}} = \left( \bigotimes_{i=1}^{p} \boldsymbol{w_i^\intercal} \right)^{\otimes r} \cdot \tilde{\boldsymbol{X}} \cdot \tilde{\boldsymbol{H}} \cdot \tilde{\boldsymbol{X}}^\dagger \cdot \left( \bigotimes_{i=p}^{1} \boldsymbol{w_i} \right)^{\otimes r} + 1, \tag{36}$$

where

$$\boldsymbol{w_i} \equiv \begin{pmatrix} \cos\left(\theta_i\right) \\ \sin\left(\theta_i\right) \end{pmatrix} \tag{37}$$

and $\tilde{\boldsymbol{X}} \in \mathbb{C}^{2^q \times 2^n}$ is a random matrix with rows

$$\left\langle \tilde{X} \right|_{\boldsymbol{\gamma}} \equiv \langle \psi_0 | \, C^\dagger Q_{\boldsymbol{\gamma}}^\dagger. \tag{38}$$

We will proceed as follows. First, we will bound the difference in the first two moments of equation 36 and its equivalent, where the rows of $\tilde{X}$ are i.i.d. Haar random, to be exponentially small in $n$. As Haar random vectors have first three moments matching those of random Gaussian vectors (scaled by $2^{-\frac{n}{2}}$), this gives the desired convergence through second moments. Then, we will show that the characteristic functions at $x$ of equation 36 and its i.i.d. Haar random equivalent converge exponentially quickly in $n$ for small enough $x$, giving a convergence in distribution at a rate $\tilde{\Omega} \left( \frac{\lg(A)f(n)n}{A} \right)$ by Ushakov (2011). Finally, convergence in distribution to equation 31 will follow as the error in the relevant higher-order moments between Haar random and scaled Gaussian vectors exponentially decays in $n$ by a generalization of Borel's lemma (Jiang, 2005).

Obviously the first moment of equation 36 matches that of the i.i.d. Haar random case; off-diagonal entries in the path integral average to zero, and the diagonal entries are correct as $C$ is drawn from a unitary 2-design (DiVincenzo et al., 2002). Let us now consider the second moments of the nontrivial parts of both, where we are concerned with terms of the form:

$$c_{\alpha\beta\mu\nu} = \mathbb{E} \left[ \langle \psi_0 | \, C^\dagger Q_{\gamma_{\alpha}}^\dagger H Q_{\gamma_{\beta}} C \, | \psi_0 \rangle \, \langle \psi_0 | \, C^\dagger Q_{\gamma_{\mu}}^\dagger H Q_{\gamma_{\nu}} C \, | \psi_0 \rangle \right], \tag{39}$$

and how they differ from the i.i.d. Haar random equivalent

$$h_{\alpha\beta\mu\nu} = \mathbb{E} \left[ \langle \psi_0 | \, U_{\alpha}^\dagger H U_{\beta} \, | \psi_0 \rangle \, \langle \psi_0 | \, U_{\mu}^\dagger H U_{\nu} \, | \psi_0 \rangle \right]. \tag{40}$$

First, assume $\alpha = \beta = \mu = \nu$; as $C$ is drawn from a unitary 2-design (DiVincenzo et al., 2002), the terms are equal. Similarly, if

$$\boldsymbol{\gamma_{\alpha}} \oplus \boldsymbol{\gamma_{\beta}} \oplus \boldsymbol{\gamma_{\mu}} \oplus \boldsymbol{\gamma_{\nu}} \neq 0, \tag{41}$$

then both expectations are equal to zero; this is because $c_{\alpha\beta\mu\nu}$ must have an odd number of some $Q$, and $h_{\alpha\beta\mu\nu}$ an odd number of some $U$ (or $U^\dagger$).

Let us now consider when the above conditions are not satisfied. We consider simultaneously terms of the form

$$\left( \langle \psi_0 | \, C^\dagger Q_{\gamma_{\alpha}}^\dagger R Q_{\gamma_{\beta}} C \, | \psi_0 \rangle + \gamma_{\alpha j} \leftrightarrow \gamma_{\beta j} \right) \left( \langle \psi_0 | \, C^\dagger Q_{\gamma_{\mu}}^\dagger R' Q_{\gamma_{\nu}} C \, | \psi_0 \rangle + \gamma_{\mu j} \leftrightarrow \gamma_{\nu j} \right), \tag{42}$$

i.e. all terms summed where unequal components of $\boldsymbol{\gamma_{\alpha}}$ and $\boldsymbol{\gamma_{\beta}}$ (and $\boldsymbol{\gamma_{\mu}}$ and $\boldsymbol{\gamma_{\nu}}$) are swapped. Note that the parity of the permutation determines the sign of the term in equation 36 (and thus in equation 42). Here, $R$ and $R'$ are terms in the Pauli expansion of $\tilde{H}$. Consider the largest $j$ where $\boldsymbol{\gamma_{\alpha}}$ and $\boldsymbol{\gamma_{\beta}}$ differ; consider the sum of each pair of terms in equation 42 that have component $j$

permuted, but are equal at all $k < j$. Each pair of terms is of the form (with relative signs made explicit)

$$
\langle \psi_0 | \, C^\dagger A Q_j A' R B' B C \, | \psi_0 \rangle - \langle \psi_0 | \, C^\dagger A A' R B' Q_j B C \, | \psi_0 \rangle
$$
$$
= 2 \langle \psi_0 | \, C^\dagger A Q_j A' R B' B C \, | \psi_0 \rangle \, \mathbf{1}_{[Q_j, A' R B'] \neq \mathbf{0}}
\tag{43}
$$

for some $A, A', B, B'$. For all $Q_j$ that commute with $A' R B'$, the two terms cancel. In particular, $Q_j A' R B'$ cannot be proportional to the identity. As $\tilde{H}$ is traceless, both $R$ and $R'$ are also not proportional to the identity. This can be done inductively for all $j$ where $\gamma_\alpha$ and $\gamma_\beta$ differ.

Consider the case where $\gamma_\alpha + \gamma_\beta \neq \gamma_\mu + \gamma_\nu$; we must have that $\gamma_\alpha$ and $\gamma_\beta$ have a coordinate $i$ where they are both one, and where $\gamma_\mu$ and $\gamma_\nu$ are both zero (assuming equation 41 is not satisfied). By equation 43, WLOG we can consider the product of Pauli observables between the two $Q_i$ as being not proportional to the identity. Then, averaging over $Q_i$ will yield zero. This is the same as the i.i.d. Haar random case, as every term in the expansion of equation 42 must have only one of some unitary when $\gamma_\alpha + \gamma_\beta \neq \gamma_\mu + \gamma_\nu$.

Finally, consider the case where $\gamma_\alpha + \gamma_\beta = \gamma_\mu + \gamma_\nu$. Under this constraint, we must have the same number of terms in each sum in equation 42; we call this number of terms $2^c$. In the Pauli case, every time we combine terms as in equation 43 introduces an overall factor of $4$, and we average only over the anticommuting Pauli operators. As the value of the expectation over $C$ is independent of the (nonidentity) Pauli in the expectation value, this introduces a factor of $\frac{1}{2}$ every time we combine terms. This gives

$$
2^c \mathbb{E}_{C \sim \mathbb{C}_n} \left[ \langle \psi_0 | \, C^\dagger S C \, | \psi_0 \rangle \langle \psi_0 | \, C^\dagger S' C \, | \psi_0 \rangle \right],
\tag{44}
$$

for some $S$ and $S'$ that are equal if and only if $R = R'$. Similarly, in the i.i.d. Haar random case, only products of terms with $\gamma_\alpha = \gamma_\mu$ and $\gamma_\beta = \gamma_\nu$ are homogeneous in their unitaries and give nonzero expectations, yielding

$$
2^c \mathbb{E}_{U \sim \mathbb{C}_n} \left[ \langle \psi_0 | \, U_\alpha^\dagger R U_\beta \, | \psi_0 \rangle \langle \psi_0 | \, U_\beta^\dagger R' U_\alpha \, | \psi_0 \rangle \right].
\tag{45}
$$

If $R \neq R'$ (and $S \neq S'$), these are both zero. If $R = R'$ (and $S = S'$), the latter is equal to $2^{c-n}$ and the former to $2^{c-n} \left( 1 + \mathrm{O} \left( 2^{-n} \right) \right)$. Putting everything together and explicitly writing the overall factor of $\frac{2^n}{\| H - \lambda_1 \|_*}$, we have that the error in the second moment is on the order of

$$
\epsilon_2 = \frac{2^{2n}}{\| H - \lambda_1 \|_*^2} \left( 2^{-n} \sqrt{\sum_{i=1}^{A} \alpha_i^2} \right)^2,
\tag{46}
$$

where $\alpha_i$ are the coefficients of the Pauli expansion of $H - \overline{\lambda}$. We also have that

$$
\sum_{i=1}^{A} \alpha_i^2 = 2^{-n} \left\| H - \overline{\lambda} \right\|_{\mathrm{F}}^2 = m^{-1} 2^{-n} \left\| H - \lambda_1 \right\|_*^2,
\tag{47}
$$

where $m$ is defined as in equation 26. Thus,

$$
\epsilon_2 = 2^{-(n + \lg(m))}.
\tag{48}
$$

Let us now consider the $t$th moment for $t \geq 3$. We will bound the higher moments of both models, and show that their characteristic functions have infinite radii of convergence. Then, by showing that the difference in these characteristic functions vanishes exponentially in $n$ for all $x \geq 0$ bounded below $\frac{\lg(A) f(n) n}{A}$, we will show that the two models converge in distribution at a rate $\tilde{\Omega} \left( \frac{\lg(A) f(n) n}{A} \right)$.

By grouping terms as in equation 42, it is sufficient to only bound

$$
b_t = \left( \overline{\lambda} - \lambda_1 \right)^{-t} \mathbb{E}_{C \sim \mathbb{C}_n} \left[ \prod_{i=1}^{t} \left( \sum_{j=1}^{A} \alpha_j \langle \psi_0 | \, C^\dagger S_{ij} C \, | \psi_0 \rangle \right) \right],
\tag{49}
$$

where $S_{ij}$ is not proportional to the identity, $S_{ij} \neq S_{i'j}$ for all $i \neq i'$, and $A$ is the number of terms in the Pauli decomposition of $\tilde{H}$. If a term in the expansion of equation 49 contains two $S_{ij}$ that

anticommute, the contribution to the moment from that term is zero as $C \left| \psi_0 \right\rangle$ is a stabilizer state for all $C$. Generally, the contribution to the moment is maximized when the $S_{ij}$ are "maximally dependent"—that is, for $d$ distinct $S_{ij}$ in a term, the contribution to the moment is maximized when the $S_{ij}$ are generated by a cardinality $\lfloor \lg(d) + 1 \rfloor$ subset of them. Thus, the contribution to the moment is bounded by $2^{-c \lfloor \lg(d) + 1 \rfloor n}$ for some constant $c$ (Aaronson & Gottesman, 2004). Note that this also bounds the i.i.d. Haar random case. Putting everything together and using the multinomial theorem, the $t$th moment of the nontrivial part of both distributions is bounded by

$$b_t \leq \sum_{\sum_i k_i = t} 2^{-c \lfloor \lg(\|\boldsymbol{k}\|_0) + 1 \rfloor n} \binom{t}{k_1, \ldots, k_A} \prod_{i=1}^{A} \left( \frac{\alpha_i}{\bar{\lambda} - \lambda_1} \right)^{k_i}. \tag{50}$$

This corresponds to the case where $S_{ij} = S_{ij'} \equiv S_i$, i.e. when there is maximal dependence between the matrix elements. Here, $k_i$ indexes how many times $S_i$ appears in a term in equation 49, and $\|\cdot\|_0$ denotes the number of nonzero coordinates of $\cdot$. By equation 32, as $t \to \infty$ for any given $A$ and $n$,

$$\frac{b_t}{t!} \leq (1 + \mathrm{o}(1)) \, 2^{-t \lg(t) - \frac{1}{2} \lg(2\pi t) + t \lg\left( \frac{\mathrm{e}A}{f(n)} \right) - c \lg(A) n}. \tag{51}$$

Thus, the Taylor series of the characteristic functions of both distributions have infinite radii of convergence, and both are completely determined by their moments. Furthermore, equation 51 gives us that the difference in their characteristic functions at $0 \leq x < \frac{c \lg(A) f(n) n}{A}$ is on the order of $\exp\left( \frac{Ax}{f(n)} - c \lg(A) n \right)$ as $n \to \infty$. As the two distributions have equal moments for $t \leq 2$, it can then be shown (Ushakov, 2011) that the error in Lévy distance between the two is $\tilde{\mathrm{O}}\left( \left( \frac{\lg(A) f(n) n}{A} \right)^{-1} \right)$. $\qquad \square$

Now that we have shown the weak convergence of our random class of VQAs to a random field on the hypertorus, we can combine this result with a multidimensional generalization of the Welch–Satterthwaite equation (Satterthwaite, 1946; Welch, 1947) to show that our distribution of VQAs has first two moments matching that of a Wishart hypertoroidal random field (WHRF). Once again, under further assumptions on the spectrum of $H$ we will also bound the higher moments of the two distributions to show convergence in distribution.

**Theorem 5** (XHX RFs as WHRFs). *The random field given by equation 31 has first two moments equal to the Wishart hypertoroidal random field (WHRF)*

$$F_{WHRF}(\boldsymbol{\theta}) = m^{-1} \sum_{i_1, \ldots, i_r, i_1', \ldots, i_r' = 1}^{2^p} w_{i_1} \ldots w_{i_r} J_{i_1, \ldots, i_r, i_1', \ldots, i_r'} w_{i_1'} \ldots w_{i_r'}, \tag{52}$$

*where $\boldsymbol{J} \sim \mathcal{CW}_{2^q}(m, \boldsymbol{I}_{2^q})$ is a complex Wishart random matrix and the* effective degrees of freedom *defined in equation 26 is formally a real number, but can be rounded to the nearest natural number with negligible error. Furthermore, assuming the largest eigenvalue of $\tilde{H}$ as defined in equation 29 is at most $2^{cn}$ for some constant $c$ bounded below 1, their distributions are bounded in Lévy distance by $2^{-\Omega(\min(n, \lg(m)))}$.*

*Proof.* By the unitary invariance of random matrices with Gaussian entries, by diagonalizing $\tilde{H}$ we can rewrite $F_{\mathrm{XHX}}$ as the random field

$$F_{\mathrm{XHX}}(\boldsymbol{w}) = \|H - \lambda_1\|_*^{-1} \left( \bigotimes_{i=1}^{p} \boldsymbol{w_i} \right)^{\otimes r} \cdot \sum_{i=1}^{2^n} \left( (h_i - \bar{\lambda}) \boldsymbol{X_i} \otimes (\boldsymbol{X_i})^\dagger + \bar{\lambda} - \lambda_1 \right) \cdot \left( \bigotimes_{i=p}^{1} \boldsymbol{w_i} \right)^{\otimes r}, \tag{53}$$

where $\boldsymbol{X_i}$ is the $i$th column of $\boldsymbol{X}$ and $h_i$ are the eigenvalues of $H$. The sum over Kronecker products of columns is just the weighted sum of (at most) $2^n$ independent Wishart random variables, each with a single degree of freedom. It is known (Khuri et al., 1994; 2011; Pivaro et al., 2017) that the first two moments of this weighted sum of independent Wishart random variables is equal (up to rounding of the degrees of freedom) to that of the single Wishart random variable

$$\boldsymbol{J} \sim \mathcal{CW}_{2^q}(m, m^{-1} \boldsymbol{I}_{2^q}), \tag{54}$$

where $m$ is defined as in equation 26.

Let us now consider higher moments of both distributions. A useful property of both $F_{\text{XHX}}$ and $F_{\text{WHRF}}$ is that they are invariant under rotations on the hypertorus $\boldsymbol{w} \mapsto \boldsymbol{O} \cdot \boldsymbol{w}$ (for real orthogonal $\boldsymbol{O} \in \text{SO}(2)^{\otimes p}$) due to the invariance of the Wishart distribution under orthogonal transformations (Eaton, 2007). Due to this property, we will often take

$$\boldsymbol{w} = \boldsymbol{n} \equiv (1, 0, \ldots, 0)^{\mathsf{T}}, \tag{55}$$

i.e. perform calculations at a fixed point $\boldsymbol{\theta} = \boldsymbol{0}$ on the hypertorus. For instance, by inspection of the marginal distributions of the elements of $\boldsymbol{X} \otimes \boldsymbol{X}^{\dagger}$ and $\boldsymbol{J}$ (Srivastava, 2003; Yu et al., 2014), we immediately see that

$$\left( \bigotimes_{i=1}^{p} \boldsymbol{w_i} \right)^{\otimes r} \cdot \left( \boldsymbol{X} \otimes \boldsymbol{X}^{\dagger} \right) \cdot \left( \bigotimes_{i=p}^{1} \boldsymbol{w_i} \right)^{\otimes r} \sim \Gamma(1, 1) \tag{56}$$

and

$$F_{\text{WHRF}}(\boldsymbol{w}) \sim m^{-1} J_{(1,\ldots,1),(1,\ldots,1)} \sim \Gamma\left(m, m^{-1}\right); \tag{57}$$

here, $\Gamma(k, \theta)$ is a gamma distributed random variable with shape $k$ and scale $\theta$. We therefore have that the moment-generating function for $F_{\text{XHX}}(\boldsymbol{w})$ is

$$M_{\text{XHX}}(x) = \mathrm{e}^{x} \prod_{i=1}^{2^n} \left( 1 - \frac{h_i - \overline{\lambda}}{\|H - \lambda_1\|_*} x \right)^{-1} = \mathrm{e}^{x} \det\left( 1 - 2^{-n} \tilde{H} x \right)^{-1} \tag{58}$$

and for $F_{\text{WHRF}}(\boldsymbol{w})$ is

$$M_{\text{WHRF}}(x) = \left( 1 - \frac{x}{m} \right)^{-m}. \tag{59}$$

Assuming the largest eigenvalue of $\tilde{H}$ is at most $2^{cn}$, we see that these moment generating functions differ at any given $0 \le x < 2^{\min((1-c)n, \lg(m))}$ by at most $\mathrm{O}\left( 2^{-3(1-c)n} x^3 + m^{-3} x^3 \right)$. As the two distributions have equal first and second moments, it can then be shown (Ushakov, 2011) that the error in Lévy distance between the two is bounded by $2^{-\Omega(\min(n, \lg(m)))}$. $\qquad \square$

Combining the two theorems, we roughly see that under reasonable assumptions on the spectrum of $H$ the random fields induced by the specific class of VQAs we consider can be approximated by WHRFs up to an error on the order of $\tilde{\mathrm{O}}\left( \left( \frac{\lg(A) f(n) n}{A} \right)^{-1} + m^{-1} \right)$ as $m, n \to \infty$.

## A.3 DISCUSSION OF THE MAPPING

Let us now briefly discuss the intuition and assumptions behind the results proved in Appendix A.2, beginning with the random class of ansatzes we consider. Of course, in practice, VQA ansatzes are not chosen at random. Indeed, VQA ansatzes have a layered structure that precludes any independence between layers even if the layers were randomly chosen. Though this randomness assumption is strong, heuristically deep enough circuits (that are independent of the problem Hamiltonian) will still look roughly uniform over stabilizer states in the Feynman path integral expansion performed in the proof of Theorem 4, giving qualitatively similar results. Furthermore, though throughout this paper we consider results in expectation over this distribution of ansatzes, we find numerically in Sec. 5.1 that our analytic results seem to also hold in distribution; we therefore suspect that our analytic results in Appendix C hold more generally for individual ansatzes that are independent of the problem Hamiltonian.

Given the randomized class of ansatzes, in Theorem 4 we show that the VQA loss function is close in distribution to that of the random field given in equation 31 (the "XHX" model). Intuitively, this just stems from the fact that different paths in the Feynman path integral are matrix elements in uniformly random stabilizer states. We then show that the error induced in higher moments by taking each of these paths to be independent vanishes as $n \to \infty$. To prove this formally, we rely on the boundedness of equation 32 to bound higher moments of the distribution. Luckily, in practice this bound holds; for extensive Hamiltonians, one expects $f(n) \gtrsim n$.

Theorem 5 extends Theorem 4 by showing that the XHX model can be written as a sum of Wishart models weighted by the (scaled and shifted) eigenvalues of $H$, which can then be approximated by a single Wishart model. Heuristically, one can think of complex Wishart matrices as multidimensional generalizations of the gamma distribution; then, the approximation used in Theorem 5 is just a multidimensional generalization of the Welch–Satterthwaite approximation (Satterthwaite, 1946; Welch, 1947). This approximation (in both the univariate and multivariate cases) is exact in the first two moments of the distribution when the *effective degrees of freedom* $m$ given in equation 26 is allowed to be real. In practice, $m$ is rounded to the nearest natural number, inducing a slight error in the approximation. Generally, errors in higher moments in the Welch–Satterthwaite approximation may be large when the moments of the approximated distribution is large, particularly when the coefficients of the sum can have arbitrary sign and are at different scales (Satterthwaite, 1946; Khuri, 1994). However, for physically relevant Hamiltonians, the spectral radius is much smaller than $2^n$, and the coefficients of the sum are approximately equal. We show that under such conditions, errors in the moment generating functions vanish as $m, n \to \infty$ at the given rate.

The effective degrees of freedom $m$ as in equation 26 can be interpreted as roughly a signal-to-noise ratio of the mean eigenvalue of $H - \lambda_1$, and generically is at least exponentially large in $n$ (for small eigenvalue spacings, as is typically found in physical Hamiltonians studied with VQAs (Peruzzo et al., 2014; Farhi et al., 2014; Romero et al., 2018)). We show in Sec. 4.3.1 that $m$ sharply dictates the variational loss landscape; for a number of independent parameters $p \geq 2m$, local minima concentrate near the global minimum. Conversely, for $p$ bounded below $2m$, local minima concentrate far away from the global minimum. This would imply that for the class of randomized ansatz we consider here, training large instances is infeasible. However, consider an ansatz that is allowed to depend on the problem instance $H$, such as in the Hamiltonian variational ansatz (HVA) (Wecker et al., 2015). With a clever enough ansatz, one can in principle "reweigh" the coefficients of equation 53 by having a nonuniform distribution over stabilizer states in the Feynman path integral expansion of equation 36, effectively making $m$ smaller. This would be consistent with what was numerically investigated in prior work (Wiersema et al., 2020) (and in Sec. 5.2), where it was shown that even for a modest number of parameters the distribution of local minima concentrate near the global minimum for the HVA. We leave further investigation in this direction for future work.

Finally, we note that all of our asymptotic equivalence results so far have been shown to converge at a rate

$$\rho \equiv \lg(A) f(n) n / A, \tag{60}$$

which is typically $\gtrsim \lg(n)$ for physically relevant (i.e. two-local with arbitrary range, molecular in the plane wave dual basis (Babbush et al., 2018), etc.) Hamiltonians. In Appendix D, we give the loss landscape of WHRFs (equation 52) as $p, m \to \infty$, taking into account large deviations in $p$. If $p$ grows as $\Omega(\lg(\rho))$, then in principle uncontrolled large deviations in the convergence of VQAs to WHRFs will dominate the asymptotics of the landscape (equation 18). In particular, with probability $\sim \rho^{-1}$, deviations of the eigenvalues of the Hessian on the order of the eigenvalues themselves can occur, which are then "blown up" by a factor exponentially large in $p$ if all deviations constructively interfere. Thus, though equation 18 holds for WHRFs, it does not necessarily hold for VQAs when $p = \Omega(\lg(\rho))$. If the deviations of eigenvalues of the Hessian due to the mapping from VQAs to WHRFs are roughly independent between eigenvalues, however, then these deviations are further exponentially suppressed in $p$, and the result holds independently of how $p$ scales with $n$. We believe in practice this is what occurs, and see numerically in Sec. 5.1 that our analytic results hold well even when $p \gg n$.

## B  THE KAC–RICE FORMULA AND ITS ASSUMPTIONS

For completeness, we state the formal version of Lemma 1—with all assumptions—here. We borrow heavily from (Adler & Taylor, 2009). By $\nabla f$, we mean the covariant gradient of $f$.

**Lemma 3** (Kac–Rice formula (Adler & Taylor, 2009)). *Let $M$ be a compact, oriented, $N$-dimensional $C^1$ manifold with $C^1$ Riemannian metric $g$. Let $B \subset \mathbb{R}^K$ be an open set such that $\partial B$ has dimension $K - 1$. Let $f : M \to \mathbb{R}^K$ be a random field on $M$, and let $\iota(\cdot)$ denote the index of $\cdot$. Furthermore, assume that:*

> *1. All components of $f$, $\nabla f$, and $\nabla^2 f$ are almost surely continuous and have finite variances over $M$.*

2. *The marginal density $p_t \left( \nabla f \left( t \right) \right)$ of $\nabla f$ at $t \in M$ is continuous at $\nabla f = \mathbf{0}$.*

3. *The conditional densities $p_t \left( \nabla f \left( t \right) \mid f \left( t \right), \nabla^2 f \left( t \right) \right)$ are bounded above and continuous at $\nabla f = \mathbf{0}$, uniformly in $t \in M$.*

4. *The conditional densities $p_t \left( \det \left( \nabla^2 f \left( t \right) \right) \mid \nabla f \left( t \right) = \mathbf{0} \right)$ are continuous in the neighborhood of $\det \left( \nabla^2 f \right) = 0$ and $\nabla f \left( t \right) = \mathbf{0}$, uniformly in $t \in M$.*

5. *The conditional densities $p_t \left( f \left( t \right) \mid \nabla f \left( t \right) = \mathbf{0} \right)$ are continuous for all $f$ and for all $\nabla f$ in a neighborhood of $\mathbf{0}$, uniformly in $t \in M$.*

6. *The Hessian moments are bounded, i.e.*

$$\sup_{t \in M} \max_{i,j} \mathbb{E} \left[ \left| \left( \nabla^2 f \left( t \right) \right)_{i,j} \right|^N \right] < \infty. \tag{61}$$

7. *The moduli of continuity with respect to (the canonical metric induced by) $g$ of each component of $f$, $\nabla f$, and $\nabla^2 f$ all satisfy*

$$\mathbb{P} \left[ \omega \left( \eta \right) > \epsilon \right] = \mathrm{o} \left( \eta^N \right) \tag{62}$$

*for all $\epsilon > 0$ as $\eta \to 0^+$.*

*Then,*

$$\mathbb{E} \left[ \mathrm{Crt}_k^f \left( B \right) \right] = \int_M p_{\boldsymbol{\sigma}} \left( \nabla f \left( \boldsymbol{\sigma} \right) = 0 \right) \tag{63}$$
$$\times \mathbb{E} \left[ \left| \det \left( \nabla^2 f \left( \boldsymbol{\sigma} \right) \right) \right| \mathbf{1} \left\{ f \left( \boldsymbol{\sigma} \right) \in B \right\} \mathbf{1} \left\{ \iota \left( \nabla^2 f \left( \boldsymbol{\sigma} \right) \right) \leq k \right\} \mid \nabla f \left( \boldsymbol{\sigma} \right) = 0 \right] \mathrm{d} \boldsymbol{\sigma},$$

*where $\mathrm{d} \boldsymbol{\sigma}$ is the volume element induced by $g$ on $M$.*

It is obvious by Lemma 4 that conditions 2-6 are satisfied by WHRFs given $B = (0, u)$. Furthermore, as $F$ is a polynomial in $\{\cos \left( \theta_i \right), \sin \left( \theta_i \right)\}$, $F$ and its derivatives are continuous for any value of the components of $m^{-1} \boldsymbol{J}$, and all have finite variance. Similarly, it is easy to see that the modulus of continuity of $f$ and its gradients go as $J \eta^r$ as $\eta \to 0^+$, where $J$ is the largest component of $m^{-1} \boldsymbol{J}$. As the distributions of the components of a Wishart matrix have exponential tails, the probability that $J = \Omega \left( \eta^{-r} \right)$ is indeed $\mathrm{o} \left( \eta^N \right)$ and therefore all conditions are satisfied by WHRFs.

## C  THE LOSS LANDSCAPE OF WISHART HYPERTOROIDAL RANDOM FIELDS

### C.1  THE JOINT DISTRIBUTION OF $F_{\mathrm{WHRF}}$ AND ITS DERIVATIVES

In order to utilize the Kac–Rice formula (Lemma 3), we must calculate the joint distribution of the random field

$$F_{\mathrm{WHRF}} \left( \boldsymbol{\theta} \right) = m^{-1} \sum_{i_1, \ldots, i_r, i_1', \ldots, i_r' = 1}^{2^p} w_{i_1} \ldots w_{i_r} J_{i_1, \ldots, i_r, i_1', \ldots, i_r'} w_{i_1'} \ldots w_{i_r'}, \tag{64}$$

with its derivatives. In the course of proving Theorem 5, we already have shown that the function value is gamma distributed (see equation 57). Here, we explicitly calculate the distribution of the Hessian when given the function value and that the covariant gradient is zero, and also calculate the distribution of the gradient given the function value. We will heavily lean on the rotational invariance property of the distribution discussed in the proof of Theorem 5, with $\boldsymbol{n}$ once again the fixed point with all $\boldsymbol{\theta_i} = 0$. Note that for the given embedding of the hypertorus into $\mathbb{R}^{2^p}$, the Christoffel symbols are zero (i.e. we are considering the Euclidean hypertorus) and thus for the most part we can ignore the distinction between covariant and normal derivatives. Here, we choose local coordinates $\boldsymbol{\theta}$ such that:

$$\boldsymbol{w_i} = \begin{pmatrix} \cos \left( \theta_i \right) \\ \sin \left( \theta_i \right) \end{pmatrix}. \tag{65}$$

Perhaps surprisingly, we will find that conditioned on being at a critical point at a specified energy, the Hessian takes the simple form of a normalized and shifted Wishart matrix summed with a normalized GOE matrix. The gradient conditioned on the function value is similarly simple, given by independent Gaussian variables.

**Lemma 4** (Hessian and gradient distributions). *The scaled Hessian $m\partial_i\partial_j F_{WHRF}(\boldsymbol{w})$ conditioned on $F_{WHRF}(\boldsymbol{w}) = x$ and $\partial_k F_{WHRF}(\boldsymbol{w}) = 0$ is distributed as*

$$m\tilde{C}_{ij}(x) = -2rmx\delta_{ij} + rW_{ij} + r\sqrt{2mx}N_{ij}, \tag{66}$$

*where $\boldsymbol{W} \sim \mathcal{W}_p(2m, \boldsymbol{I_p})$ and $\boldsymbol{N} \sim GOE_p$ are independent. Furthermore, the scaled gradient $m\partial_k F_{WHRF}(\boldsymbol{w})$ conditioned on $F_{WHRF}(\boldsymbol{w}) = x$ is distributed as*

$$m\tilde{G}_k(x) = \sqrt{2mrx}N_k, \tag{67}$$

*where $N_k$ are i.i.d. standard normally distributed random variables independent from all $W_{ij}$ and $N_{ij}$.*

*Proof.* Without loss of generality we take $\boldsymbol{w} = \boldsymbol{n}$. Let $\boldsymbol{i} \in \{1,2\}^{\times p}$ be the vector with the $i$th component equal to 2 and all others equal to 1, $(\boldsymbol{i},\boldsymbol{j})$ similar with both the $i$th and $j$th component, and $\boldsymbol{b}$ the vector with all components equal to 1. Taking derivatives explicitly yields

$$m\partial_i F_{\text{WHRF}}(\boldsymbol{n}) = 2\operatorname{Re}\left\{J_{(\boldsymbol{i},\boldsymbol{b},\dots,\boldsymbol{b}),(\boldsymbol{b},\dots,\boldsymbol{b})}\right\} + \dots + 2\operatorname{Re}\left\{J_{(\boldsymbol{b},\dots,\boldsymbol{i}),(\boldsymbol{b},\dots,\boldsymbol{b})}\right\} \tag{68}$$

and

$$m\partial_i\partial_j F_{\text{WHRF}}(\boldsymbol{n}) = -2r\delta_{ij}J_{(\boldsymbol{b},\dots,\boldsymbol{b}),(\boldsymbol{b},\dots,\boldsymbol{b})}$$
$$+ 2\operatorname{Re}\left\{J_{(\boldsymbol{i},\boldsymbol{b},\dots,\boldsymbol{b}),(\boldsymbol{j},\boldsymbol{b},\dots,\boldsymbol{b})}\right\} + 2\operatorname{Re}\left\{J_{(\boldsymbol{i},\boldsymbol{b},\dots,\boldsymbol{b}),(\boldsymbol{b},\boldsymbol{j},\dots,\boldsymbol{b})}\right\} + \dots + 2\operatorname{Re}\left\{J_{(\boldsymbol{b},\dots,\boldsymbol{b},\boldsymbol{i}),(\boldsymbol{b},\dots,\boldsymbol{b},\boldsymbol{j})}\right\}$$
$$+ 2\operatorname{Re}\left\{J_{((\boldsymbol{i},\boldsymbol{j}),\boldsymbol{b},\dots,\boldsymbol{b}),(\boldsymbol{b},\dots,\boldsymbol{b})}\right\} + 2\operatorname{Re}\left\{J_{(\boldsymbol{i},\boldsymbol{j},\dots,\boldsymbol{b}),(\boldsymbol{b},\dots,\boldsymbol{b})}\right\} + \dots + 2\operatorname{Re}\left\{J_{(\boldsymbol{b},\dots,\boldsymbol{b},(\boldsymbol{i},\boldsymbol{j})),(\boldsymbol{b},\dots,\boldsymbol{b})}\right\}. \tag{69}$$

As $\boldsymbol{J}$ is a Wishart matrix with identity scale matrix, it can be written as $\boldsymbol{X} \cdot \boldsymbol{X}^\dagger$ for $\boldsymbol{X}$ a $2^q \times m$ matrix with i.i.d. standard complex normal entries. By performing an LQ decomposition of $\boldsymbol{X}$, one can then by inspection determine the distributions of the entries of $\boldsymbol{J}$ (Srivastava, 2003; Yu et al., 2014). For ease of notation, we let $\tau : \{1,2\}^{\times q} \to \{1,\dots,2^q\}$ be a mapping between representations of the indices of $J$, with the convention $\tau((\boldsymbol{b},\dots,\boldsymbol{b})) = 1$. We then find (taking $\tau((\boldsymbol{i},\dots,2)) < \tau((\boldsymbol{j},\dots,2))$ WLOG) that

$$2J_{(\boldsymbol{b},\dots,\boldsymbol{b}),(\boldsymbol{b},\dots,\boldsymbol{b})} = 2mF_{\text{WHRF}}(\boldsymbol{n}), \tag{70}$$

$$2\operatorname{Re}\left\{J_{(\boldsymbol{i},\dots,\boldsymbol{b}),(\boldsymbol{b},\dots,\boldsymbol{b})}\right\} = \sqrt{2mF_{\text{WHRF}}(\boldsymbol{n})}M_{(\boldsymbol{b},\dots,\boldsymbol{b}),(\boldsymbol{i},\dots,\boldsymbol{b})}, \tag{71}$$

$$2\operatorname{Re}\left\{J_{(\boldsymbol{i},\boldsymbol{j},\dots,\boldsymbol{b}),(\boldsymbol{b},\dots,\boldsymbol{b})}\right\} = \sqrt{2mF_{\text{WHRF}}(\boldsymbol{n})}M_{(\boldsymbol{b},\dots,\boldsymbol{b}),(\boldsymbol{i},\boldsymbol{j},\dots,\boldsymbol{b})}; \tag{72}$$

and, for $\tau((\boldsymbol{i},\dots,\boldsymbol{b})) \leq m$,

$$2\operatorname{Re}\left\{J_{(\boldsymbol{i},\dots,\boldsymbol{b}),(\boldsymbol{j},\dots,\boldsymbol{b})}\right\} = \sqrt{2\Gamma_{(\boldsymbol{i},\dots,\boldsymbol{b})}}M_{(\boldsymbol{i},\dots,\boldsymbol{b}),(\boldsymbol{j},\dots,\boldsymbol{b})}$$
$$+ \sum_{\mu=1}^{\tau((\boldsymbol{i},\dots,\boldsymbol{b}))-1} M_{\tau^{-1}(\mu),(\boldsymbol{i},\dots,\boldsymbol{b})}M_{\tau^{-1}(\mu),(\boldsymbol{j},\dots,\boldsymbol{b})} + \sum_{\mu=1}^{\tau((\boldsymbol{i},\dots,\boldsymbol{b}))-1} \tilde{M}_{\tau^{-1}(\mu),(\boldsymbol{i},\dots,\boldsymbol{b})}\tilde{M}_{\tau^{-1}(\mu),(\boldsymbol{j},\dots,\boldsymbol{b})} \tag{73}$$

and otherwise

$$2\operatorname{Re}\left\{J_{(\boldsymbol{i},\dots,\boldsymbol{b}),(\boldsymbol{j},\dots,\boldsymbol{b})}\right\} = \sum_{\mu=1}^{m} M_{\tau^{-1}(\mu),(\boldsymbol{i},\dots,\boldsymbol{b})}M_{\tau^{-1}(\mu),(\boldsymbol{j},\dots,\boldsymbol{b})}$$
$$+ \sum_{\mu=1}^{m} \tilde{M}_{\tau^{-1}(\mu),(\boldsymbol{i},\dots,\boldsymbol{b})}\tilde{M}_{\tau^{-1}(\mu),(\boldsymbol{j},\dots,\boldsymbol{b})}. \tag{74}$$

Here, $\boldsymbol{M}$ and $\tilde{\boldsymbol{M}}$ are symmetric with off-diagonal entries i.i.d. drawn from the standard normal distribution, and $\Gamma_{\tau^{-1}(\mu)} \equiv M_{\tau^{-1}(\mu),\tau^{-1}(\mu)}^2$ has entries i.i.d. drawn from $\Gamma(m-\mu+1,1)$. Note that each $\sqrt{2\Gamma}$ is chi-square distributed with $2(m-\mu+1)$ degrees of freedom; therefore, equation 73 and equation 74 can be considered as elements of a real Wishart matrix $\tilde{\boldsymbol{W}}$ with $2m$ degrees

of freedom. Also, note that equation 73 and equation 74 are independent of $\partial_k F_{\text{WHRF}}(\boldsymbol{n})$ when conditioned on $F_{\text{WHRF}}(\boldsymbol{n}) \equiv x = 0$. If $x \neq 0$, the condition $\partial_k F_{\text{WHRF}}(\boldsymbol{n}) = 0$ is equivalent to taking each sum over the elements of $\boldsymbol{M}$ from $\mu = 2$ instead of $\mu = 1$, which is equivalent to taking the convention $\tau((\boldsymbol{b}, \ldots, \boldsymbol{b})) = 2^q$ and shifting the indices of $\boldsymbol{M}$ and $\tilde{\boldsymbol{M}}$. Therefore, the (scaled) Hessian conditioned on $F_{\text{WHRF}}(\boldsymbol{n}) = x$ and $\partial_k F_{\text{WHRF}}(\boldsymbol{n}) = 0$ is distributed as

$$m\tilde{C}_{ij}(x) = -2rmx\delta_{ij} + \left(\boldsymbol{O} \cdot \tilde{\boldsymbol{W}} \cdot \boldsymbol{O}^{\mathsf{T}}\right)_{ij} + r\sqrt{2mx}N_{ij}; \tag{75}$$

here, $\boldsymbol{N} \sim GOE_p$ (with the convention that diagonal entries $\sim \mathcal{N}(0,2)$ and off-diagonal entries $\sim \mathcal{N}(0,1)$), and $\boldsymbol{O}$ is a matrix such that $O_{i\mu} = 1$ if and only if $\tau^{-1}(\mu)$ is of the form $(\boldsymbol{b}, \ldots, \boldsymbol{i}, \ldots, \boldsymbol{b})$, and is otherwise equal to 0. The invariance of the Wishart distribution under orthogonal transformations and partitioning (Eaton, 2007; Srivastava, 2003) leads to the final result. $\qquad \square$

### C.2 THE EXACT DISTRIBUTION OF CRITICAL POINTS

Given the joint distribution of $F_{\text{WHRF}}$, its gradient, and its Hessian, we are now equipped to calculate the expected number of critical points of a given index $k$ using the Kac–Rice formula (Lemma 3).

**Theorem 6** (Distribution of critical points in WHRFs). *Let*

$$\mu_{\boldsymbol{C}(x)} = \frac{1}{p}\sum_{i=1}^{p}\delta\left(\lambda_i^{\boldsymbol{C}(x)}\right) \tag{76}$$

*be the empirical spectral measure of the random matrix*

$$\boldsymbol{C}(x) = \frac{r}{m}\left(\boldsymbol{W} + \sqrt{2mx}\boldsymbol{N}\right), \tag{77}$$

*where $\boldsymbol{W} \sim \mathcal{W}_p(2m, \boldsymbol{I_p})$ and $\boldsymbol{N} \sim GOE_p$ are independent and $\lambda_i^{\boldsymbol{C}}(x)$ is the ith smallest eigenvalue of $\boldsymbol{C}(x)$. Then, the distribution of the expected number of critical points of index $k$ at an energy $E > 0$ of $F_{\text{WHRF}}$ is given by*

$$\mathbb{E}\left[\text{Crt}_k(E)\right]$$
$$= \left(\frac{\pi}{r}\right)^{\frac{p}{2}}\Gamma(m)^{-1}m^{(1+\gamma)m}\mathbb{E}_{\boldsymbol{C}(E)}\left[e^{p\int\ln(|\lambda - 2rE|)\mathrm{d}\mu_{\boldsymbol{C}(E)}}\mathbf{1}\left\{\lambda_{k+1}^{\boldsymbol{C}(E)} \geq 2rE\right\}\right]E^{(1-\gamma)m-1}e^{-mE}, \tag{78}$$

*where*

$$\gamma = \frac{p}{2m}. \tag{79}$$

*Proof.* As discussed in Appendix B, the assumptions of the Kac–Rice formula (i.e. Lemma 3) are satisfied. Furthermore, due to the invariance of the Wishart distribution with respect to rotations on the hypertorus (Eaton, 2007; Srivastava, 2003), we can integrate out the volume element independently; the volume of $(S^1)^{\times p}$ is

$$\int_{(S^1)^{\times p}}\mathrm{d}\boldsymbol{w} = (2\pi)^p. \tag{80}$$

Additionally, we have from Lemma 4 that the probability density of the gradient vector being zero at any $\boldsymbol{w}$ conditioned on $H_{\text{WHRF}}(\boldsymbol{w}) = x$ is

$$p_{\boldsymbol{w}}\left(\boldsymbol{\nabla}H_{\text{WHRF}}(\boldsymbol{w}) = 0 \mid H_{\text{WHRF}}(\boldsymbol{w}) = x\right) = \left(\frac{4\pi rx}{m}\right)^{-\frac{p}{2}}. \tag{81}$$

Taking the expectation over $x$ via equation 57 and using the Hessian distrribution from Lemma 4, we have from Lemma 1 that

$$\mathbb{E}\left[\text{Crt}_k(B = (0, E))\right] = \left(\frac{\pi}{r}\right)^{\frac{p}{2}}\Gamma(m)^{-1}m^{(1+\gamma)m}$$
$$\times \int_0^E \mathbb{E}_{\boldsymbol{C}(x)}\left[e^{p\int\ln(|\lambda - 2rx|)\mathrm{d}\mu_{\boldsymbol{C}(x)}}\mathbf{1}\left\{\lambda_{k+1}^{\boldsymbol{C}(x)} \geq 2rx\right\}\right]x^{(1-\gamma)m-1}e^{-mx}\mathrm{d}x. \tag{82}$$

Taking the derivative of this cumulative distribution with respect to $E$ yields the final result. $\qquad \square$

## D   LOGARITHMIC ASYMPTOTICS VIA FREE PROBABILITY THEORY

Though equation 78 is exact, it is difficult to use in practice. Luckily, we are able to use a surprising fact about the eigenvalue distributions of Wishart and GOE matrices; asymptotically, the empirical spectral distributions of these matrices weakly converge to fixed distributions. Concretely, in the limit $p \to \infty$ where $\gamma = \frac{p}{2m}$ is held constant, the eigenvalue distribution of $\boldsymbol{W}/2m$ where $\boldsymbol{W} \sim \mathcal{W}_p(2m, \boldsymbol{I_p})$ weakly converges to the *Marchenko–Pastur distribution* (Marčenko & Pastur, 1967):

$$\mathrm{d}\mu_{\text{M.P.}} = \left(1 - \gamma^{-1}\right) \mathbf{1}\left\{\gamma > 1\right\} \delta\left(\lambda\right) \mathrm{d}\lambda + \frac{1}{2\pi\gamma\lambda}\sqrt{\left(\left(1 + \sqrt{\gamma}\right)^2 - \lambda\right)\left(\lambda - \left(1 - \sqrt{\gamma}\right)^2\right)}\,\mathrm{d}\lambda. \quad (83)$$

Similarly, the eigenvalue distribution of $\boldsymbol{N}/\sqrt{p}$ where $\boldsymbol{N} \sim \text{GOE}_p$ weakly converges to the *Wigner semicircle distribution* (Wigner, 1958):

$$\mathrm{d}\mu_{\text{s.c.}} = \frac{1}{2\pi}\sqrt{4 - \lambda^2}\,\mathrm{d}\lambda. \quad (84)$$

Furthermore, by using free probability theory one can find the asymptotic distribution of eigenvalues for a weighted sum of these matrices, given their eigenbases are in "generic position" with respect to each other. We now give a brief review of free probability theory—at least in the context of random matrix theory—here. Later, we will also briefly review large deviations theory, which we use to bound the probability of large deviations from the weak convergence of the eigenvalue distributions of Wishart and GOE matrices. Note that, as we are unable to control large deviations in Theorem 4, in principle large deviations in the weak convergence of VQAs to WHRFs could dominate the large deviations in WHRFs; however, as discussed in Appendix A.3, this provably does not occur at shallow enough depths with respect to $n$, and there are reasons to believe it does not occur even at large depths (which we additionally give numerical evidence for in Sec. 5).

We begin by reviewing the techniques in free probability theory and large deviations theory that we use in studying the asymptotic behavior of equation 78.

### D.1   FREE PROBABILITY THEORY

*Free probability theory* is the study of noncommutative random variables. Specializing to random matrix theory on $N \times N$ matrices, we define the unital linear functional

$$\phi\left(X\right) \equiv \frac{1}{N}\mathbb{E}\left[\text{tr}\left(X\right)\right] \quad (85)$$

as the free analogue of the expectation. Note that the eigenvalues of a matrix $A$ are completely constrained by the trace of powers $A^k$—therefore, one can study the average distribution of the eigenvalues of a random matrix $A$ via the moments $\phi\left(A^k\right)$. *Free independence* (or *freeness*) is a generalization of the notion of independence in commutative probability theory to free probability theory. In the context of random matrix theory, two $N \times N$ random matrices $A$ and $B$ are said to be freely independent if the mixed moments are identically zero; that is,

$$\phi\left(\left(A^{m_1} - \phi\left(A^{m_1}\right)\right)\left(B^{n_1} - \phi\left(B^{n_1}\right)\right)\ldots\left(A^{m_k} - \phi\left(A^{m_k}\right)\right)\left(B^{n_k} - \phi\left(B^{n_k}\right)\right)\right) = 0 \quad (86)$$

for all $n_i, m_i \in \mathbb{N}$. Roughly, the free independence of two random matrices means that their eigenbases are in "generic position" from one another.

Taking the analogy with commutative probability theory further, the analogue of the moment-generating function associated with the distribution of a random variable is the *Stieltjes transform* of the measure $\mu$:

$$G_\mu\left(z\right) = \int \frac{\mathrm{d}\mu\left(t\right)}{z - t}, \quad (87)$$

which can be inverted via the Stieltjes inversion formula:

$$\mathrm{d}\mu\left(t\right) = -\frac{1}{\pi}\lim_{\epsilon \to 0^+} \text{Im}\left\{G_\mu\left(t + \mathrm{i}\epsilon\right)\right\}\mathrm{d}t. \quad (88)$$

Similarly, the free analogue of the cumulant-generating function is the *R-transform*, which can be defined via the Stieltjes transform as the solution to the implicit equation:

$$\mathcal{R}_\mu\left(G_\mu\left(z\right)\right) + \frac{1}{G_\mu\left(z\right)} = z. \quad (89)$$

The $R$-transform is important in that, if two random variables $A$ and $B$ are freely independent with probability measures $\mu_A$ and $\mu_B$ respectively, the probability measure $\mu_{A+B}$ of $A + B$ satisfies

$$\mathcal{R}_{\mu_{A+B}} = \mathcal{R}_{\mu_A} + \mathcal{R}_{\mu_B}. \tag{90}$$

This can be interpreted as the free analog of the additivity of cumulants for commutative random variables. The probability measure $\mu_{A+B}$ is called the *free convolution* of $\mu_A$ and $\mu_B$, and is denoted using the notation

$$\mu_{A+B} = \mu_A \boxplus \mu_B. \tag{91}$$

Thus, given the probability distributions of two free random variables $A$ and $B$, there is a prescription for determining the probability distribution of their sum by taking their free convolution, just as the convolution in commutative probability theory describes the distribution of the sum of random variables.

### D.2 LARGE DEVIATIONS THEORY

In order to bound the probability of large deviations from the weak convergence of the eigenvalue distribution of $C$ to its asymptotic limit we will use results from large deviations theory, which we briefly review here. A sequence of measures $\{\mu_n\}$ is said to satisfy a large deviation principle in the limit $n \to \infty$ with speed $s(n)$ and lower semicontinuous rate function $I$ with codomain $[0, \infty]$ if and only if (Dembo & Zeitouni, 2010)

$$-\inf_{x \in \Gamma^\circ} I(x) \leq \lim\inf_{n \to \infty} \frac{1}{s(n)} \ln\left(\mu_n\left(\Gamma\right)\right) \leq \lim\sup_{n \to \infty} \frac{1}{s(n)} \ln\left(\mu_n\left(\Gamma\right)\right) \leq -\inf_{x \in \overline{\Gamma}} I(x) \tag{92}$$

for all Borel measurable sets $\Gamma$ that all $\mu_n$ are defined on. Here, $\overline{\Gamma}$ denotes the closure of $\Gamma$ and $\Gamma^\circ$ the interior of $\Gamma$. The rate function $I$ is said to be *good* if all level sets of $I$ are compact. Large deviations theory will be useful for us to bound the probabilities of large deviations of the empirical spectral distribution of $\mu_{C(x)}$ as $p \to \infty$, and show that they do not contribute to leading order in the (logarithmic) asymptotic distribution of critical points. We do this using Varadhan's lemma, which we state now.

**Lemma 5** (Varadhan's lemma (Dembo & Zeitouni, 2010)). *Suppose $\{\mu_n\}$ satisfies a large deviation principle with speed $s(n)$ and good rate function $I$ and let $\phi$ be a real-valued continuous function. Further assume either the* tail condition

$$\lim_{M \to \infty} \lim\sup_{n \to \infty} \frac{1}{s(n)} \ln\left(\mathbb{E}_{X_n \sim \mu_n}\left[e^{s(n)\phi(X_n)} \mathbf{1}\left\{\phi\left(X_n\right) \geq M\right\}\right]\right) = -\infty, \tag{93}$$

*or the* moment condition *for some $\gamma > 1$*

$$\lim\sup_{n \to \infty} \frac{1}{s(n)} \ln\left(\mathbb{E}_{X_n \sim \mu_n}\left[e^{\gamma s(n)\phi(X_n)}\right]\right) < \infty. \tag{94}$$

*Then,*

$$\lim_{n \to \infty} \frac{1}{s(n)} \ln\left(\mathbb{E}_{X_n \sim \mu_n}\left[e^{s(n)\phi(X_n)}\right]\right) = \sup_x \left(\phi\left(x\right) - I\left(x\right)\right). \tag{95}$$

### D.3 LOGARITHMIC ASYMPTOTICS OF THE DISTRIBUTION OF CRITICAL POINTS

Equipped with these mathematical tools, we prove our first result on the asymptotic behavior of $\mu_{C(x)}$, which is present in the expectation of equation 78.

**Lemma 6** (Asymptotic behavior of $\mu_{C(x)}$). *Define $G_x^*(z)$ as the implicit solution of the equation*

$$8r^3\gamma^2 x G_x^*(z)^3 - 2r\gamma(z + 2rx)G_x^*(z)^2 + (z - 2r(1 - \gamma))G_x^*(z) - 1 = 0 \tag{96}$$

*with the smallest imaginary part. Define*

$$\mathrm{d}\mu_x^* \equiv -\frac{1}{\pi}\operatorname{Im}\{G_x^*\}\,\mathrm{d}\lambda. \tag{97}$$

*Let $p, m \to \infty$ as $\gamma = \frac{p}{2m}$ is held constant. Then, the empirical spectral measure $\mu_{C(x)}$ satisfies a large deviation principle as $p \to \infty$ with speed $p^2$ with good rate function uniquely minimized at $\mu_x^*$ with a value of $0$.*

*Proof.* The empirical spectral measure of the random matrix $N/\sqrt{p}$ satisfies a large deviation principle at a scale $p^2$, with good rate function minimized by Wigner's semicircle law (Arous & Guionnet, 1997). Similarly, the empirical spectral measure of the random matrix $W/2m$ satisfies a large deviation principle at a scale $p^2$, with good rate function minimized by the Marchenko–Pastur distribution (Hiai & Petz, 1998). As the $R$-transform of the empirical spectral distribution of $A$ satisfies the scaling property

$$\mathcal{R}_{aA}(z) = a\mathcal{R}_A(az), \tag{98}$$

the $R$-transform of the empirical spectral distribution of the weighted GOE term of $C$ is of the form

$$\mathcal{R}_{\text{GOE}}(z) = 4r^2\gamma xz \tag{99}$$

and the $R$-transform of the weighted Wishart term is of the form

$$\mathcal{R}_{\text{Wishart}}(z) = \frac{2r}{1 - 2r\gamma z}. \tag{100}$$

By the asymptotic freeness of independent GOE and Wishart matrices (Hiai & Petz, 2006), $\mu_{C(x)}$ converges weakly to the fixed measure $\mu^*$ with $R$-transform

$$\mathcal{R}_x(z) = \mathcal{R}_{\text{Wishart}}(z) + \mathcal{R}_{\text{GOE}}(z). \tag{101}$$

Equation 96 and equation 97 now follow from inverting the $R$-transform $\mathcal{R}_x$ via equation 89 and equation 88, respectively.

We now consider large deviations in the weak convergence $\mu_{C(x)} \rightsquigarrow \mu_x^*$. Conditioning on the empirical spectral distribution of $W/2m$ and using the "strongest growth wins" principle (Dembo & Zeitouni, 2010), we have that $\mu_{C(x)}$ satisfies a large deviation principle with speed $p^2$ with rate function given by

$$I(\mu) = \inf_{\mu_{W/2m}} \left( J_{\mu_{W/2m}}(\mu) + K\left(\mu_{W/2m}\right) \right); \tag{102}$$

here, $K$ is the rate function governing convergence of the empirical spectral distribution of the Wishart ensemble (Hiai & Petz, 1998) and $J_{\mu_{W/2m}}$ is the rate function governing convergence of the empirical spectral distribution of a fixed matrix with asymptotic eigenvalue distribution $\mu_{W/2m}$ summed with a GOE matrix (Guionnet & Zeitouni, 2002). This sum is obviously uniquely minimized by $\mu = \mu^*$, when $I(\mu^*) = 0$. □

Now, we examine the asymptotic behavior of the smallest eigenvalue $\lambda_1^{C(x)}$ of $C(x)$. Unlike the empirical spectral measure $\mu_{C(x)}$ which satisfies a large deviation principle at a speed $p^2$, we will see that this eigenvalue satisfies a large deviation principle at a speed $p$, with deviations at this speed to the left of the asymptotic value $\lambda_{x,1}^*$.

**Lemma 7** (Asymptotic behavior of $\lambda_1^{C(x)}$). *Let $\lambda_{x,1}^*$ be the infimum of the support of $\mu_x^*$ as defined in equation 97. Then, the smallest eigenvalue $\lambda_1^{C(x)}$ of $C(x)$ satisfies a large deviation principle with speed $p$ with good rate function that is infinite at $y > \lambda_{x,1}^*$ and is uniquely minimized at $y = \lambda_{x,1}^*$ with a value of $0$.*

*Proof.* The limiting smallest eigenvalues $\lambda_1^W, \lambda_1^N$ of $W/2m$ and $N/\sqrt{p}$ both satisfy large deviation principles with speed $p$ that are infinite for $\lambda_1$ in the bulk of their respective limiting empirical spectral distributions (Johansson, 2000; Arous et al., 2001). As in the proof of Lemma 6, we condition on large deviations of these eigenvalues (Dembo & Zeitouni, 2010) and therefore have that the rate function governing $\lambda_1^{C(x)}$ is

$$I(y) = \inf_{\lambda_1^W, \lambda_1^N} \left( J_{\lambda_1^W, \lambda_1^N}(y) + K\left(\lambda_1^W\right) + L\left(\lambda_1^N\right) \right); \tag{103}$$

here, $K$ is the rate function governing the convergence of $\lambda_1^W$, $L$ that of $\lambda_1^N$, and $J$ that of the smallest eigenvalue $C(x)$ conditioned on the eigenvalue distributions of $W$ and $N$. Using known results on the large deviations of the smallest eigenvalue of the sum of two matrices with fixed eigenvalues (i.e. $J_{\lambda_1^W, \lambda_1^N}$) (Guionnet & Maïda, 2020), we see that $I(y)$ is infinite for $y > \lambda_{x,1}^*$ and is uniquely minimized at $y = \lambda_{x,1}^*$ with a value of $0$. □

Using Lemmas 6 and 7, we can prove the following logarithmic asymptotics on the expectation term in equation 78. We will find that neither the large deviations in the convergence $\mu_{C(x)}$ or $\lambda_1^{C(x)}$ will contribute to leading order in the logarithmic asymptotics of $\mathrm{Crt}_k(E)$, as at a speed $p$ the only large deviations are $\lambda_1^{C(x)} \leq \lambda_{x,1}^*$ which are dominated by $\lambda_1^{C(x)} = \lambda_{x,1}^*$ in the expectation.

**Lemma 8** (Logarithmic asymptotics of the determinant). *Let* $\mathrm{d}\mu_E^*$ *be the spectral measure given in equation 97, with* $\lambda_{E,1}^*$ *the infimum of its support. Let* $p, m \gg 1$ *with* $\frac{p}{2m} = \gamma = \mathrm{O}(1)$. *Then,*

$$
\begin{aligned}
&\frac{1}{p} \ln \left( \mathbb{E}_{\boldsymbol{C}(E)} \left[ e^{p \int \ln(|\lambda - 2rE|) \mathrm{d}\mu_{\boldsymbol{C}(E)}} \mathbf{1} \left\{ \lambda_{k+1}^{\boldsymbol{C}(E)} \geq 2rE \right\} \right] \right) \\
&= \int \ln \left( \mathbf{1} \left\{ \lambda_{E,1}^* \geq 2rE \right\} |\lambda - 2rE| \right) \mathrm{d}\mu_E^* + \mathrm{o}(1).
\end{aligned}
\tag{104}
$$

*Proof.* As $\mu_{\boldsymbol{C}(E)}$ satisfies a large deviation principle with speed $p^2$ with rate function minimized at $\mu_E^*$ by Lemma 6, and as $\mathbf{1}\left\{ \lambda_1^{\boldsymbol{C}(E)} \geq 2rE \right\} \leq 1$, we have that the tail condition of Varadhan's lemma at speed $p$ is satisfied (Dembo & Zeitouni, 2010) and therefore

$$
\begin{aligned}
&\lim_{p \to \infty} \frac{1}{p} \ln \left( \mathbb{E}_{\boldsymbol{C}(E)} \left[ e^{p \int \ln(|\lambda - 2rE|) \mathrm{d}\mu_{\boldsymbol{C}(E)}} \mathbf{1} \left\{ \lambda_1^{\boldsymbol{C}(E)} \geq 2rE \right\} \right] \right) \\
&= \sup_{\lambda \in \mathbb{R}} \left( \int \ln \left( \mathbf{1} \left\{ \lambda \geq 2rE \right\} |\lambda - 2rE| \right) \mathrm{d}\mu_E^* - I(\lambda) \right).
\end{aligned}
\tag{105}
$$

Here, $I$ is as in equation 103. The supremum over $\lambda$ is obviously achieved when $\lambda = \lambda_{E,1}^*$ by the properties of $I$ discussed in Lemma 7, giving the leading order term in equation 104. The result being exact in the $p \to \infty$ limit gives the subleading $\mathrm{o}(1)$. $\square$

Using Lemma 8, we can therefore finally calculate the logarithmic asymptotic distribution of local minima of a WHRF.

**Theorem 7** (Logarithmic asymptotics of the local minima distribution). *Let* $\mathrm{d}\mu_E^*$ *be the spectral measure given in equation 97, with* $\lambda_{E,1}^*$ *the infimum of its support. Let* $p, m \gg 1$ *with* $\frac{p}{2m} = \gamma = \mathrm{O}(1)$. *Then, the expected distribution of local minima of* $F_{WHRF}$ *at a fixed energy* $E > 0$ *is given by*

$$
\begin{aligned}
\frac{1}{p} \ln \left( \mathbb{E}\left[ \mathrm{Crt}_0(E) \right] \right) &= \frac{1}{2} \ln \left( \frac{\pi q}{2\gamma} \right) + \frac{1}{2\gamma}(1 - E) + \frac{1}{2}\left( \gamma^{-1} - 1 \right) \ln(E) \\
&+ \int \ln \left( \left| \frac{\lambda}{r} - 2E \right| \mathbf{1} \left\{ \frac{\lambda_{E,1}^*}{r} \geq 2E \right\} \right) \mathrm{d}\mu_E^* + \mathrm{o}(1).
\end{aligned}
\tag{106}
$$

*Proof.* The result follows directly from applying Lemma 8 to Theorem 6. $\square$

Note that, though we only prove the asymptotic distribution of local minima in Theorem 7, we expect similar theorems to also hold for critical points of constant index $k$ (taking $\lambda_{E,1}^* \mapsto \lambda_{E,k}^*$ in the integrand). The only difference in the derivation is the exact form of the large deviations of the $k$th smallest eigenvalue of $\boldsymbol{C}(x)$. This is similar to the case in Gaussian hyperspherical random fields (Auffinger et al., 2013).

# E   DETAILS OF THE NUMERICAL SIMULATIONS

We now give further details on the numerical simulations performed in Sec. 5. We performed all simulations via Qiskit (Abraham et al., 2019), and used standard gradient descent (via the method of finite differences) to optimize the VQA loss function

$$
F(\boldsymbol{\theta}) = \langle \boldsymbol{\theta} | H_{\boldsymbol{T}, \boldsymbol{U}} | \boldsymbol{\theta} \rangle
\tag{107}
$$

until convergence. $H_{T,U}$ is the 1D $n$ site *spinless Fermi–Hubbard Hamiltonian* (Negele & Orland, 1998)

$$H_{\boldsymbol{T},\boldsymbol{U}} = -\sum_{i=1}^{n-1} T_i \left( c_i^\dagger c_{i+1} + c_{i+1}^\dagger c_i \right) + \sum_{i=1}^{n-1} U_i c_i^\dagger c_i c_{i+1}^\dagger c_{i+1}, \tag{108}$$

where $c$ is the fermionic annihilation operator. $T_i$ and $U_i$ are i.i.d. normally distributed in order to break translational invariance. In our simulations, these random variables were centered at $T = 1$ and $U = 2$, respectively, and each had a variance of $10^{-2}$.

Our implementation of gradient descent used a learning rate of $0.05$ and a momentum of $0.9$, and halted when either the function value improved by no more than $10^{-5}$ or after $10^6$ iterations, whichever came first. We initialized each instance at a uniformly random point in parameter space, with each parameter initialized within $[-2\pi, 2\pi]$.

To estimate the empirical distribution of local minima for the studied instances of the varitional quantum eigensolver (VQE) (Peruzzo et al., 2014), we repeated this procedure 52 times, using a new ansatz and uniformly random starting point for each training instance. We also verified numerically that $m$ as defined in equation 26 is at least on the order of $2^n$ for $H_{1,2}$, though we directly used equation 26 when computing $\gamma$. In all plotted instances, we normalize the energy scale by a factor of $c_{\text{VQA}}$, where

$$c_{\text{VQA}} = \overline{\lambda} - \lambda_1; \tag{109}$$

this is just the overall factor of equation 8. These units are such that the mean eigenvalue of $H - \lambda_1$ in the subspace of interest is at $E = 1$.

In Sec. 5.2, we tested our analytic results against a Hamiltonian informed ansatz. Specifically, we used the *Hamiltonian variational ansatz* (HVA) (Wecker et al., 2015). For the Fermi–Hubbard Hamiltonian of equation 108, each HVA layer is of the form

$$U_i^{\boldsymbol{T},\boldsymbol{U}} \left( \boldsymbol{\theta_i} \right) = \mathrm{e}^{-\mathrm{i}\theta i, 2 H_{T,U,\text{odd}}} \mathrm{e}^{-\mathrm{i}\theta i, 2 H_{T,U,\text{even}}} \mathrm{e}^{-\mathrm{i}\theta_{i,1} H_{T,U,\text{Coulomb}}}. \tag{110}$$

Here, $H_{\boldsymbol{T},\boldsymbol{U},\text{Coulomb}}$ is composed of the terms proportional to $U_i$ in $H_{\boldsymbol{T},\boldsymbol{U}}$, $H_{\boldsymbol{T},\boldsymbol{U},\text{even}}$ the hopping terms on even links, and $H_{\boldsymbol{T},\boldsymbol{U},\text{odd}}$ the hopping terms on odd links. We took the starting state $|\psi_0\rangle$ to be the computational basis state $|1\rangle$ on the first $\frac{n}{2}$ qubits and $|0\rangle$ on the other $\frac{n}{2}$ qubits. To observe the effects of scaling the number of independent parameters $p$, we overparameterize our ansatz at a fixed overall depth by fixing the total number of ansatz layers $U_i^{\boldsymbol{T},\boldsymbol{U}}$ to be 6, but introduce extra parameters to govern each evolution. For instance, for a multiplicative factor $f = 2$, we double the number of parameters by splitting into a sum of two terms

$$H_{\boldsymbol{T},\boldsymbol{U},\text{Coulomb}} = H_{\boldsymbol{T},\boldsymbol{U},\text{Coulomb}}^{(1)} + H_{\boldsymbol{T},\boldsymbol{U},\text{Coulomb}}^{(2)}, \tag{111}$$

$$H_{\boldsymbol{T},\boldsymbol{U},\text{even}} = H_{\boldsymbol{T},\boldsymbol{U},\text{even}}^{(1)} + H_{\boldsymbol{T},\boldsymbol{U},\text{even}}^{(2)}, \tag{112}$$

$$H_{\boldsymbol{T},\boldsymbol{U},\text{odd}} = H_{\boldsymbol{T},\boldsymbol{U},\text{odd}}^{(1)} + H_{\boldsymbol{T},\boldsymbol{U},\text{odd}}^{(2)}, \tag{113}$$

and parameterize the evolution under each term separately. For $f = 1$, this ansatz preserves the fermion number of the initial state; thus, for these simulations we calculate $m$ in this $\frac{n}{2}$-fermion subspace. For large $f$, this parameterization breaks the fermion number conservation of the ansatz, but still preserves the parity of the fermion number. In practice, then, the $\gamma$ we compute should be considered an upper bound on the true $\gamma$, strengthening our empirical results.

