# OpenReview forum: "Critical Points in Quantum Generative Models"
_ICLR.cc/2022/Conference — ICLR 2022 Poster_

### Official Review · Reviewer_F6sD · 2021-11-01

**Correctness:** 4
**Technical Novelty And Significance:** 4
**Empirical Novelty And Significance:** 3
**Recommendation:** 8
**Confidence:** 3

**Main Review:**

The main results are stated and proven in theorems 1,2 and 3.  Theorem 1 defines a randomized set of quantum variational problems and asserts that their loss landscape is the same as that of a random potential model with potential given by a complex Wishart random matrix.  Theorem 2 then computes the distribution of critical points of this random potential model, while theorem 3 gives its asymptotics.   This result is then interpreted to demonstrate the phase transition discussed in the summary.
In section 5 a numerical confirmation of the results is done on a 1d Hubbard Hamiltonian.
It is also suggested that a similar algorithm but using operators adapted to the Hamiltonian might do significantly better, and this suggestion is supported with citations and another numerical experiment.

I found the main question well motivated and clearly asked, and an impressive variety of techniques were brought to bear on it.
I was surprised by the basic claim that this class of models is sufficiently random to justify dropping all of the higher moments in computing the ground state energy.  The arguments given for this are explicit but intricate and while they look reasonable, I did not check them in complete detail.  I suspect that there is a much simpler argument waiting to be found, perhaps building on the comments in Appendix A.3.
Given this claim, the rest is conceptually straightforward but rather nontrivial to carry through successfully.

**Summary Of The Paper:**

This paper obtains rather impressive results about the problem of estimating a quantum ground state energy using a variational method which could be implemented on a quantum computer.  Given a Hamiltonian H, the method generates quantum states by applying a succession of n-qubit Pauli operators to a random initial state (both chosen without knowledge of H), and measure the expectation of H.
Previous work suggested that there is a critical (and exponentially large) number of parameters required to get a good estimate.
Here a theorem is formulated and proven that there is a phase transition of this type, with good estimates only above a threshold in the number of parameters, and with an explicit expression for the threshold.  The proof uses a combination of physics methods (series expansions), random matrix theory and the Kac-Rice method to count critical points of a random function.

**Summary Of The Review:**

A sharp formulation and sound theoretical analysis of a well motivated and important claim, that quantum variational states which do not depend on specifics of the Hamiltonian under study, require exponentially many parameters to produce accurate ground state energies.

---

> ### Author Response · Authors · 2021-11-10
> **Review Response**
>
> We thank the reviewer for taking the time to read the manuscript, and for providing insightful and kind remarks. We also suspect that there may be a simpler (or at least more physically motivated) way to prove Theorem 1 without having to resort to the technical aspects of Appendix A.2---we hope in future work to develop techniques along these lines that will also work for Hamiltonian informed models.

---

### Official Review · Reviewer_zxWF · 2021-11-02

**Correctness:** 2
**Technical Novelty And Significance:** 3
**Empirical Novelty And Significance:** 2
**Recommendation:** 6
**Confidence:** 4

**Main Review:**

### Pros

The work leverages the concepts of random matrix theory to analyze the optimization
performance of quantum generative models.

### Cons

1. The paper is really hard to read. Several English grammar and word mistakes prevent
the readers from clearly understanding the main idea in the paper.

2. A few concepts introduced in this work are either incorrect or imprecise and detailed descriptions of quantum generative circuits
are totally missing.

3. The proposed methods are not very technically sound because the barren plateau associated
with the optimization of QNN mainly arises from the quantum noise of NISQ devices.
But the work does not consider the noisy quantum circuits, which makes the work
insignificant in practice.

4. Terminologies are introduced for granted without any explanation.

5. The numerical experiments are not linked to the quantum simulation at all, and the
experimental simulation cannot corroborate the quantum theorems.

6. I believe the paper including the abstract needs a major rewritten.

## Feedbacks

1. The abstract and each section contain some space that could be improved.

For example,

(1) the claim “The clustering of local minima of the loss function near the global minimum” is imprecise
because the local optimal minima are found but not clustered;

(2) The sentence “...to local minima that are good approximations of the global minimum” is wrong because the
local minima should be close to the global one but not an approximation. (e.g., approximation means you use other functions and methods to get the result.)

2. The connections between local minima to the random matrix theory are still not very clear.

- In particular, the authors do not take into account how to make use of their theory to improve optimization performance.

3. Some key concepts related to QNN on NISQ devices are missing, e.g., barren plateau problems.

**Summary Of The Paper:**

The work attempts to connect the optimization of quantum generative models with random fields on manifolds. Some theoretical and numerical analyses are provided.

The major idea is to apply random matrix theory for variational quantum circuits. When the authors provide some numerical experiments, overall the validation is not very strong and could be improved.

The connection between applying random fields and quantum circuit training is not very direct based on the current writing. Some related references in the field are missing. (e.g., [1,2]).

***

**Reference**

1. Random matrix spectral form factor of dual-unitary quantum circuits." Communications in Mathematical Physics 387.1 (2021): 597-620.

2. Abrupt transitions in variational quantum circuit training." Physical Review A 103.3 (2021): 032607.

**Summary Of The Review:**

When the paper provides some attempts to combine random matrix to VQC training, the theoretical findings are not very correlated to their empirical findings. The current version is not very ready to get published and several claims and description need to be modified.


**Additional Comments**

1. The information on optimization settings (optimizers and initialization) is not provided in
the paper.

2. Since the quantum generative model refers to a large model family, the ‘critical points’
for each concrete one are not the same. The paper does not show what specific models are
used. Is it a quantum Bayesian network? Quantum hidden Markov model? Or anything
else?

3. Many important related papers are not cited, e.g, for the expressiveness of QNN, which has to be credited from

“The effect of data encoding on the expressive power of variational quantum machine learning models,” Physical Review A, 2021 - APS

***

### Post-Author Response

I think most of my comments have been carefully covered by the authors. Thank you for your efforts during the rebuttal period.

With the current version, some of the writing, reproducible discussion, and presentation have been largely improved.

I therefore increase my score to 6; one remaining place that would still improve is the current VQA experiments is limited to a single empirical case.

Some of the variational circuit works have been studied in a classical-quantum format. The author could consider to add some connection to enhance potential impact of the works.

---

> ### Author Response · Authors · 2021-11-10
> **Review Response: Pros and Cons**
>
> We thank the reviewer for taking the time to read the manuscript, and for the comments and concerns. To address all comments, we have split our response into two comments: one responding to the pros and cons, and another responding to the feedback.
>
> In response to the missing citations: we thank the reviewer for pointing out these works. We have added references to [2] throughout the work where appropriate. However, we struggle to see the connection between [1] and our work, as it seems [1] studies level spacings in chaotic quantum condensed matter systems. Could the reviewer elaborate on this connection?
>
> In response to poor English and grammar: we apologize, and have taken a closer look throughout the manuscript correcting such mistakes. On the subject of clarity, we have also added more discussion of our results in the introduction (Sec. 1.2), also following the comments of Reviewer 5aN5.
>
> In response to the imprecise concepts and definitions: could the reviewer clarify which concepts were incorrectly or imprecisely defined? We have added a note in the preliminaries (Sec. 2.1) explaining that there are other formulations of quantum generative modeling, and that we here choose to focus on variational quantum algorithms. We hope this clarifies that we are only considering a certain class of quantum generative models. We chose to save the introduction of variational quantum algorithms for Sec. 2.1 rather than referencing them in the abstract in an effort to make the manuscript more accessible for a general machine learning audience.
>
> In response to the question of barren plateaus: though barren plateaus were recently seen to manifest in noisy quantum circuits with depth growing linearly in the problem size, in their original formulation barren plateaus were studied in the noiseless regime (see "Barren plateaus in quantum neural network training landscapes" by McClean et al., "Cost Function Dependent Barren Plateaus in Shallow Parametrized Quantum Circuits" by Cerezo et al., etc.). In fact, our work demonstrates the computational intractability of training variational algorithms in a regime where noise-induced barren plateaus are not present: when the depth of the circuit is sublinear in the problem size (as discussed in Sec. 1.1). That being said, we do believe the study of the impact of noise on these models is important, and hope to in future work study the critical point behavior of noisy variational quantum circuits: we have added a point along these lines in Sec. 6.
>
> In response to the question of using terminology not yet introduced: we have gone through the paper and could not find any such instances, as most are defined in Sec. 2. Could the reviewer clarify and give an example of such a case?
>
> In response to the empirical results not being linked to the theoretical results: numerically, we study two instances. In Sec. 5.1, we numerically study exactly the ansatz we consider theoretically, and show the concentration of local minima on the order of the bounds arising from the simulations occuring at finite system sizes (i.e. up to deviations on the order of the square root of gamma). We have now made this more clear in this section. Though we do not study the model in Sec. 5.2 analytically, we give a heuristic explanation for this behavior in Appendix A.3 and emphasize that these numerical results are to be considered in contrast to the regime of our proofs.
>
> In response to the paper needing a major rewrite: assuming the concerns are with regard to the clarity of the work, we have gone through and cleaned up the structure of sentences, etc. We hope that this suffices in making the work more clear.

---

> ### Author Response · Authors · 2021-11-10
> **Review Response: Feedback**
>
> In response to the claim that the clustering of local minima near the global minimum is imprecise: we believe the reviewer is referring to the first sentence of our abstract. Could the reviewer clarify what is meant by "the local optimal minima are found but not clustered?" If the reviewer means with regards to e.g. parameter space instead of in function value, we have added a clarifying note in Sec. 1.1 (and throughout the paper) that here we consider the concentration of local minima in function value.
>
> In response to the objection of the use of the word "approximation": we believe the reviewer is referring to its use in the beginning of Sec. 1.1. We are unclear what the reviewer means by "approximation means you use other functions and methods to get the result," and would appreciate clarification. In the meantime, we have made more clear in Sec. 1.1 (and throughout the paper) that by "approximation," we mean in loss function value.
>
> In response to how these results can improve optimization performance: indeed, as our results show poor landscapes, we believe it is difficult to optimize the quantum models we study in depth in this work. In the final paragraph of our Conclusion (Sec. 6), we do paint a possible path forward: using Hamiltonian informed models. As discussed in Sec. 1 and Sec. 3, there exists a heuristic explanation as to why these models may perform better, that is expounded upon in more detail in Appendix A.3.
>
> In response to how these results connect with barren plateaus: we discuss this in Sec. 4.3. We also explain in Sec. 1.1 that these results hold in regimes where barren plateau results do not necessarily hold, e.g. the shallow model regime.
>
> In response to not providing details on the optimization hyperparameters for the numerics: as discussed in the Reproducibility Statement, these details are given in Appendix E. We have now added details on how we initialized the optimization procedure here as well (i.e. random initialization).
>
> In response to which quantum generative models these results apply to: as previously mentioned, we have now clarified in Sec. 2.1 that we focus on variational quantum algorithms, and not more general quantum generative models.
>
> In response to not citing the express power of QNNs: as the results the reviewer cited (and similar results on e.g. the expressive power of quantum Bayesian networks) apply to quantum generative models that are not variational quantum algorithms, we thought it outside the scope of this work to include those details here. Furthermore, in Sec. 1.1 we include a brief reference to such results in motivating the use of quantum generative models.

---

> > ### Comment · Reviewer_zxWF · 2021-11-27
> > **Re: Review Response: Feedback**
> >
> > Thank the author that carefully addresses most of the suggestions during the feedback session.
> >
> > I think most of my original concerns are on the clarity, which has been largely improved in the revised and could potentially benefit general audiences in ICLR.
> >
> > One of the remaining comments is the numerical experiments could consider different hybrid systems to potentially enlarge the impacts of the current work.
> >
> > Also, the selection of generative modeling [1] could be potentially misleading to some existing works. The authors may consider add some related discussion in their final version.
> >
> > I have improved score and recommend this paper as borderline accepted.
> >
> > ***
> >
> > 1. Benedetti, Marcello, et al. "A generative modeling approach for benchmarking and training shallow quantum circuits." npj Quantum Information 5.1 (2019): 1-9.

---

### Official Review · Reviewer_5aN5 · 2021-11-02

**Correctness:** 4
**Technical Novelty And Significance:** 4
**Empirical Novelty And Significance:** 2
**Recommendation:** 8
**Confidence:** 3

**Main Review:**

### Strengths

* The paper gives a rigorous analysis of a general result connected to an important open question in quantum machine learning, namely the ease or difficulty of finding high-quality solutions via gradient-based learning of parameterized quantum circuits. The paper's (mostly negative) result provides a useful starting point for researchers to build better quantum generative models, by avoiding the same assumptions used to derive results here.

* The writing and presentation are very clean and polished, with important results given a more accessible informal statement in the main text, along with a precise and fully rigorous statement in the appendices. This goes a long ways towards making the difficult results presented more accessible to general readers.


### Weaknesses

* Despite the quality of the writing, the paper's results are still quite technical, and likely not accessible to most readers (even those working in quantum machine learning). This isn't necessarily a fault of the paper, but rather a consequence of the high technical level needed to derive results based on a non-trivial application of random matrix theory.

* On the above note, I would encourage the authors to work a bit harder on including a more self-contained and accessible summary of their results for readers who can't or won't take the time to follow the entirety of the paper's technical development. This summary information is already available in a distributed form throughout the paper, with the description in the abstract doing a good job of qualitatively summarizing the main findings, and the last paragraph of Section 4.1 (along with Equation 16) doing a good job of defining the overparameterization factor and its defining role in this phase transition. However, it would be very useful to have a one-paragraph description in a more prominent place (end of introduction?), simply stating the expected quality of local minima as a function of (a) Hilbert space dimension (or number of qubits) and (b) quantum circuit parameter count.

**Summary Of The Paper:**

The paper analyzes a general class of variational quantum circuits using random matrix theory, and gives a theoretical characterization of the distribution of high-quality local minima of the associated Hamiltonian loss function. A type of phase transition is predicted, where the model exhibits an abundance of high-quality local minima only at very large parameter counts, and a wide gap between global and typical local minima at more typical parameter counts. These predictions are compared with small numerical experiments, as well as previous works on this topic.

**Summary Of The Review:**

Although quite technical, the paper's results are very relevant to the field of quantum machine learning, and proved in a quite general setting. I anticipate this to have a big impact on the design of variational quantum algorithms.

---

> ### Author Response · Authors · 2021-11-10
> **Review Response**
>
> We thank the reviewer for taking the time to read the manuscript, and for the very kind remarks.
>
> In response to the comment on the accessibility of the work, we have added a high level explanation of the asymptotic results at the end of the introduction (namely, in Sec. 1.2). We hope that this addition makes the main ideas of the results more clear for readers.

---

### Author Response · Authors · 2021-11-22
**Collective Review Response**

We thank all reviewers for the kind and thorough feedback.

In addition to the changes already discussed in our previous responses to reviewers, we have just submitted a revision with a few more clarity edits, in line with the suggestions of Reviewer zxWF. We hope that this aids in the readability of our work, particularly when taken in combination with the changes of our previous revision.

---

### Decision · Program_Chairs · 2022-01-20

**Decision:**

Accept (Poster)

**Comment:**

*Summary:*
 Study the location of local minima for quantum generative models.

*Strengths:*
- Rigorous analysis of an important question.
- Clear writing with important conclusions.

*Weaknesses:*
- Technical writing might not be very accessible.

*Discussion:*

Reviewers were mostly favorable about this submission. They found the topic important and the contribution significant. A main concern was that the writing might not be sufficiently self contained and the writing might not be accessible to a broad audience. Authors worked on the accessibility. In the initial review, zxWF expressed concerns about concepts, proposed methods, numerical experiments. zxWF found that the author responses carefully covered most of their comments and raised score as a consequence. zxWF still finds that some aspects could be improved, particularly in regard to experiments. F6sD found the question well motivated, the techniques impressive, and the claims important.

*Conclusion:*

Three reviewers are favorable about this work. Two of them find it good and one marginally above the acceptance threshold. I find the topic and the nature of the claims important. Considering the unanimously positive reactions from the reviewers I am recommending this article to be accepted. I ask the authors to take the comments from the reviewers carefully into account when preparing the final version of the paper.